# Combination of soil water extraction methods quantifies isotopic mixing of waters held at separate tensions in soil

William H. Bowers[1], Jason J. Mercer[1], Mark S. Pleasants[2], David G. Williams[1,2]

[1]Department of Botany, University of Wyoming, Laramie, 82070, USA

[2]Department of Ecosystem Science and Management, University of Wyoming, Laramie, 82070, USA

*Correspondence to:* William H. Bowers (willhbowers@gmail.com)

**Abstract.** Measurements of the isotopic composition of separate and potentially interacting pools of soil water provide a powerful means to precisely resolve plant water sources and quantify water residence time and connectivity among soil water regions during recharge events. Here we present an approach for quantifying the time-dependent isotopic mixing of water recovered at separate suction pressures or tensions in soil over an entire moisture release curve. We wetted oven-dried, homogenized sandy loam soil first with isotopically "light" water ($\delta^2H$ = -130‰; $\delta^{18}O$ = -17.6‰) to represent antecedent moisture held at high matric tension, and then brought the soil to near saturation with "heavy" water ($\delta^2H$ = -44‰; $\delta^{18}O$ = -7.8‰) representing new input water. Soil water samples were then sequentially extracted at three tensions ("low tension" centrifugation $\approx$ 0.016 MPa; "mid tension" centrifugation $\approx$ 1.14 MPa; and "high tension" cryogenic vacuum distillation at an estimated tension greater than 100 MPa) starting after variable equilibration periods of 0 h, 8 h, 1 d, 3 d and 7 d. We assessed differences in the isotopic composition of extracted water over the 7 d equilibration period with a MANOVA and a model quantifying time-dependent isotopic mixing of water towards equilibrium via self-diffusion. The simplified and homogenous soil structure and nearly saturated moisture conditions used in our experiment likely facilitated rapid isotope mixing and equilibration among antecedent and new input water. Despite this, the isotope composition of waters extracted at mid compared to high tension remained significantly different for up to 1 day, and that for waters extracted at low compared to high tension remained significantly different for greater than 3 days. Complete mixing (assuming no fractionation) for the pool of water extracted at high tension occurred after approximately 4.33 days. Our combination approach involving extraction of water over different domains of the moisture release curve will be useful for assessing how soil texture and other physical and chemical properties influence isotope exchange and mixing times for studies aiming to properly characterize and interpret the isotopic composition of extracted soil and plant waters, especially under variably unsaturated conditions.

## 1 Introduction

Quantifying residence time and connectivity of soil water requires methods that differentiate the isotopic signature of water pools held across different sized soil pores and ranges of matric tensions or suction pressures. A variety of field- and lab-based methods are typically employed for such analyses and each separately assesses different pools of water recovered at discrete ranges of tension (Oerter and Bowen, 2017; Orlowski et al., 2016b; Sprenger et al., 2015). These

methods effectively recover and analyze water from different soil-pore size ranges and only a few methods are capable of sampling hygroscopic water, i.e. the water that forms thin films around soil particles held at matric tensions greater than plants are able to extract. The Two Water Worlds (TWW) hypothesis (McDonnell, 2014) considers that transpiration and runoff to streams derive from separate pools of water that are incompletely mixed in time or across pore regions in the soil. Brooks et al. (2010) presented stable isotope evidence of ecohydrologic separation between plant available water in smaller pore regions and mobile water passing through preferential flow paths when smaller pores were filled, challenging the hypothesis of translatory flow and establishing a mechanism to explain the TWW hypothesis. Yet, most studies examining ecohydrologic separation and the TWW hypothesis fail to differentiate isotopic signatures beyond that of mobile water and bulk soil water. More comprehensive evaluation of soil water isotopes across multiple pore sizes and soil regions is needed to examine recharge processes explaining the TWW hypothesis (Berry et al., 2018; Brantley et al., 2017; Brooks et al., 2010; McDonnell, 2014; Sprenger et al., 2019). At a more fundamental level, such methods are needed to thoroughly address dynamics of soil water movement, mixing, and isotopic fractionation (Barnes and Allison, 1988; Braud et al., 2005; Gaj and McDonnell, 2019) to improve quantification of the water budget and trace fluxes of nutrients via water transport in the critical zone.

Characterization of water isotope ratios in soils involves careful consideration of methods used to recover soil water. Depending on the method employed, water is recovered at different energies and the proportion of water extracted is dependent on the volumetric water content of the sample and the soil water retention curve, the relationship between volumetric water content and matric potential (negative equivalent of matric tension) (Sprenger et al., 2015). Terminology for water pools recovered at different applied energies has been debated. For the purposes of relating our study to ecohydrologic separation studies, we define two commonly defined pools, gravity-drained water and matrix water, consistent with recent terminology used by Brantley et al. (2017). Gravity-drained water is the most mobile pool of water within soil that freely drains through large pores under the force of gravity. Whereas matrix water consists of capillary and hygroscopic water that does not drain freely under force of gravity but is held across a broad range of tensions by smaller pores that may or may not be accessible to plants. There is likely a continuum water mobility in soil from the largest pores to the smallest pores with progressively less water mobility as pore size decreases (Sprenger et al., 2018). However, we currently lack methodology to infer the degree of connectivity and dynamics of mixing over time between separate soil water pools extracted at different applied energies.

Methods to characterize soil water pools in situ include water vapor laser spectroscopy that assumes most mobile soil water is in equilibrium with soil water vapor (Oerter and Bowen, 2017) or field extraction using suction lysimeters (Sprenger et al., 2015). However, more often analysis of soil water isotopes involves water extraction in the lab of soil samples collected from the field. The most common of these extraction methods, in order of lowest to highest amount of energy applied to the soil sample, are suction cup lysimeters, mechanical squeezing, centrifugation, and cryogenic vacuum distillation (Sprenger et al., 2015). Suction cup lysimeters typically sample water held at low tension (0.05 to 0.10 MPa) and therefore are limited to analysis of only the highly mobile fraction of soil water, but application of much higher tensions using suction cup lysimeters is feasible (Li et al., 2007). Mechanical squeezing and centrifugation recovers water across much broader tension ranges and with no fractionation, but are unable to drain pores with diameters less than 0.03 μm (i.e. extract water held at tensions beyond 1 MPa) (Orlowski et al.,

2016b; Sprenger et al., 2015). Centrifugation is particularly useful because the rotational velocity and the centrifuge set-up are physically related to the energy applied to the soil sample and therefore the pore size drained (Edmunds and Bath, 1976). Cryogenic vacuum distillation (CVD) recovers nearly all water from a soil sample, with the more clay- and more organic-rich soil samples requiring greater extraction times or temperatures (Orlowski et al., 2016a). Each method has been used to determine the isotopic composition of specific pools of water in the soil but are rarely employed in combination to understand the dynamics of soil water pools that make up the bulk water.

CVD has been separately compared to centrifugation with the assumption that water held across matric tensions is well-mixed (Tsuruta et al., 2019), but recent findings show that applying the two methods in combination has the potential to assess water isotope compartmentation and interactions that can inform proper characterization of soil water isotopic compositions for ecohydrological studies (Adams et al., 2019). Adams et al. (2019) concluded that soil water extracted using centrifugation was consistently incompletely mixed after 72 hours of equilibration time. However, their experimental design precluded analysis of the time necessary for hygroscopic, capillary and gravitationally drained waters to completely mix. In addition to understanding mixing between water pools within soil, recent work has highlighted the importance of considering also fractionations that may affect the isotopic composition of extracted water. Isotope effects related to adhesion under various matric potentials, soil wettability, and solid interfacial chemistry of soil particles are important to consider (Gaj et al., 2019; Gaj and McDonnell, 2019).

Here we present and evaluate a step-wise procedure to recover and analyze the isotopic composition of different pools of soil water and characterize the dynamics of their interaction over time. To demonstrate the method, we confine our initial study to soil moisture conditions near saturation and investigate the time-course of mixing between waters applied sequentially to oven-dried soil. We addressed the following questions:

1. Can soil water held at different tensions be separately extracted from the same soil sample and analyzed for isotopic composition?
2. Do isotopically labeled fractions of water sequentially added to dry soil thoroughly mix?
3. Can the time-course for isotopic mixing be determined quantitatively for waters held at different tensions within soil?

## 2 Methods

### 2.1 Experimental design

Our experiment involved sequentially wetting oven-dried soil using isotopically contrasting water inputs that then allowed us to quantify the degree that separate pools of soil water mixed over time. We used a novel combination of centrifuge extraction and cryogenic vacuum distillation to recover pools of soil water held at discrete ranges of tension, spanning gravitationally drained, capillary and hygroscopic water pools. We first applied a small amount of isotopically "light" water to oven-dried soil followed by nearly saturating the soil samples with an isotopically "heavy" water. Three pools of water were recovered from wetted soils after variable mixing times through a stepwise increase of applied energy using two centrifugation speed steps followed by distilling the remaining water in the soil samples using cryogenic vacuum distillation (CVD). Subsets of samples were extracted only using CVD (hereafter called "bulk sample extraction" or "BSE") either immediately after applying only the small amount of isotopically light water

(BSE$_{light}$) or immediately after adding both the isotopically light and heavy waters (BSE$_{light+heavy}$). Prior to step-wise extraction for the remainder of the soil samples, the light and heavy water applied were allowed to freely mix and equilibrate under nearly saturated conditions for variable amounts of time: 0 hours (n=15), 8 hours (n=3), 1 day (n=3), 3 days (n=3), and 7 days (n=3). The water recovered from each soil sample, either from BSE or step-wise extractions, and from various timepoints were then analyzed for hydrogen and oxygen stable isotope ratios ($\delta^2$H and $\delta^{18}$O).

## 2.2 Experimental soil and wetting procedure

We used a sandy loam soil collected from the top 10 cm of the surface from prairie vegetation east of Laramie, WY. Soil was passed through a 2-mm sieve and all coarse litter was removed except for very fine fragments. Our experimental soil therefore was highly homogenized and lacked natural physical structure with complex soil aggregates. We employed the hydrometer method to determine soil particle size distribution using sodium hexametaphosphate as the chemical aid for dispersion (Black and Day, 1965). The particle size distribution defined by the U.S. Department of Agriculture classification system was 9% clay, 32% silt, and 59% sand. We constructed a soil retention curve (Fig. 1) using previously reported parameters for modeling water retention of sandy loam soil (van Genuchten, 1980; Kosugi et al., 2002), and highlight also the relative maximum pore size filled across the range of matric potentials as described by Schjonning (1992). We did not detect carbonates in the soil using tests with 1N HCl (Schoeneberger et al., 2012).

We prepared the homogenized soil material by oven drying a 350 g sub-sample at 105ºC for 48 hr. We then sequentially applied two isotopically distinct waters to bring the soil to near saturation. The isotopically light water used in the experiments was local tap water from the University of Wyoming campus in Laramie, and the heavy water was from multiple bottles of FIJI Water (FIJI Water LLC, Los Angeles, CA, USA). The isotope ratio value standardized to Vienna Standard Mean Ocean Water (VSMOW) for the light water was -130±2‰ for $\delta^2$H and -17.6±0.5‰ for $\delta^{18}$O (n=5) and for the heavy water was -44±2‰ for $\delta^2$H and -7.3±0.4‰ for $\delta^{18}$O (n=5). We selected these waters because of their contrasting isotopic values representing the natural range expected for cold season (light water) and warm season (heavy water) precipitation in temperate continental interior regions.

After the soil cooled from the drying procedure we applied 20ml of the light water with a spray bottle to the 350 g sub-sample and mixed by gloved hands to ensure homogenous application. 18-30g of this slightly wetted soil was gently packed to form soil columns in each of six custom made centrifuge inserts (Fig. 2). The custom steel tube inserts were perforated with small drilled holes at the bottom and fitted with a collar at the top. The collar secured the position of the insert within the sleeve at roughly 19mm above the bottom to establish a reservoir for collecting extracted water through the perforated bottom during extraction by centrifugation (below). We placed four steel screens secured by rubber o-rings at the bottom of each insert to reduce loss of soil yet permit water flow during centrifugation. In addition, we placed a small gravity secured cap on top of each insert to reduce evaporation from soil samples in inserts during equilibration and centrifugation. The caps were loose enough to not generate vacuum within the sample as water was eluted during centrifugation. We recorded weights of inserts and sleeves prior to adding the soil. Except for samples that were immediately taken for bulk sample extraction (BSE$_{light}$) using cryogenic vacuum distillation (CVD), the packed inserts were then wetted from the bottom up by immersing in a container with heavy

water at a level just below the soil level in each insert. This ensured the soil samples were wetted to near saturation by reducing the chance of air being trapped within the soil matrix. We then removed a second set of samples for bulk sample extractions (BSE$_{light+heavy}$) using CVD. The remaining samples were transferred to storage in an airtight container at 20ºC in the lab until the desired equilibration timepoints were reached. Complete saturation was not possible as some water was lost from perforations at the bottom of the inserts when they were removed from the container of heavy water. Wetted samples were weighed prior to the centrifuge extraction process to determine total wetted weight and amount of heavy water infused in each sample.

After each centrifugation step, we recorded weights of sleeves and inserts, and we collected and filtered extracted water into plastic vials with silicon caps, ready for stable isotope analysis. Vials with Parafilm were stored in a 4ºC fridge until processed. The remaining water after centrifugation was extracted using CVD ~ 2 hours to ensure all water was removed (West et al., 2006). We performed the CVD procedure at 102ºC and <0.1-2.7 Pa vacuum pressure, which were controlled and monitored using heating coils, thermistors, and vacuum gauges. The vacuum pressure used during CVD is not the same as the estimated tension applied using CVD described in section 2.3. The final sample masses after extraction were compared to oven-dried masses to determine the recovery of extracted water; every sample processed in our experiment had greater than 99% of water extracted at this step. Recent work has highlighted that CVD near 100ºC or oven drying soil near 105ºC do not extract all of the water from soil (Adams et al., 2019). The amount of water not recovered using CVD in the current study was assumed to be negligible with minimal impact on the isotopic values of extracted water.

## 2.3 Soil water extractions

We extracted water from soil using a Sorvall RC 5B Plus centrifuge fitted with a Sorvall aluminum rotor with four stainless steel sleeves designed for 50 ml Falcon Tubes (Sorvall, Newton, CT, USA). Centrifugation was performed with the cooling function activated; the internal temperature during centrifugation never exceeded 25ºC. We focused on extracting waters near two ecohydrologically relevant pressures for the waters recovered at "low" and "mid" tension: field capacity (i.e., the point at which no more water drains freely under force of gravity) and agronomic wilting point. While field capacity and wilting point varies among different soil types and plants, reference values of 0.033 MPa and 1.5 MPa for field capacity and agronomic wilting point are useful as guidelines for understanding potential boundaries on ecohydrologically separate water pools. Rotations per minute (RPM) for the centrifuge extractions at field capacity and agronomic wilting point were calculated using an equation from Nimmo et al. (1987), which relates rotational velocity to matric potential and radii of a centrifuge set-up:

$$\Psi = \rho \, \frac{\omega^2}{2} \, (r_1{}^2 - r_2{}^2) \tag{1}$$

where $\Psi$ is matric potential (Pa), $\rho$ is density of water (kg/m$^3$), $\omega$ is rotational velocity (s$^{-1}$), $r_1$ is the radius (m) from the center of the centrifuge rotor to a point of interest in the soil column during rotation, and $r_2$ is the radius from the center of centrifuge rotor to the perforated bottom of the insert where the water drains. Due to difficulties in determining the precise force distribution (Zhang et al., 2018) and since force applied using Eq. (1) varies depending

on the $r_l$ value selected, we used the center of the soil column as the point of interest for $r_l$. The first centrifuge step ("low tension") at ≈0.016 MPa was performed for three hours at 950 RPM. The second centrifuge step ("mid tension") at ≈1.14 MPa was performed for 4 hours at 8000 RPM. Afterward, the remining water in in each sample was extracted using CVD and is referenced here as "high tension" extraction; this is a fraction of water held under high tension that is rarely directly compared to more mobile waters within soils that have sufficient volumetric water content to permit sampling with methods like suction lysimeters. Applied tension using CVD is estimated to be greater than 100 MPa (Sprenger et al., 2015).

**2.4 Stable isotope analysis**

The stable isotope composition of water is expressed as δ values in units of permil (‰), where δ = ((R$_{sample}$/R$_{standard}$) − 1) × 1000. R$_{sample}$ and R$_{standard}$ are the isotope ratios of $^2$H/$^1$H or $^{18}$O/$^{16}$O for samples and those for the international standard Vienna Standard Mean Oceanic Water (VSMOW). Our measurements on samples were corrected to the VSMOW scale using working reference waters calibrated to VSMOW and SLAP reference waters obtained from the IAEA. Samples were analyzed on a Delta V isotope ratio mass spectrometer (IRMS) using a Temperature Conversion/Elemental Analyzer (TC/EA) interface (Thermo Scientific Corporation, Bremen, Germany) at the University of Wyoming Stable Isotope Facility. The analytical accuracy for the quality assessment lab reference water was 0.33‰ for δ$^2$H and 0.38‰ for δ$^{18}$O, while the analytical precision for the quality assessment lab reference water was 0.98‰ for δ$^2$H and 0.22‰ for δ$^{18}$O. We report the accuracy as the absolute difference between the mean of analyzed lab reference water samples (n=15) and the calibrated value of lab reference water. We report precision as the standard deviation of all lab reference water samples analyzed (n=15).

**2.5 Data analysis and mixing times**

We fitted a two-part mass balance mixing model using δ$^2$H and δ$^{18}$O data to account for the light and heavy water applied to the oven-dry soil and determine the distribution of added water across extracted fractions. Using Eq. (2) below, all possible combinations of replicates in this study at each equilibration timepoint and for each isotope (δ$^2$H and δ$^{18}$O) were assessed (n=54).

$$m_{LW} R_{LW} + m_{HW} R_{HW} = m_{LT} R_{LT} + m_{MT} R_{MT} + m_{HT} R_{HT} \tag{2}$$

$m$ is mass of water in kg and $R$ is isotope ratio calculated from either δ$^2$H or δ$^{18}$O values for the particular water component. The left side of Eq. (2) represents water inputs to the soil samples while the right side represents water components recovered using the step-wise extractions. To determine the percent of recovered water, the sum of outputs was divided by sum of inputs and multiplied by one hundred. Subscripts $_{HW, LT, MT,}$ and $_{HT}$ refer to the heavy water added, low tension, mid tension and high tension extracted waters. Subscript $_{LW}$ refers to light water extracted from the bulk soil extraction after only the isotopically light water was applied (BSE$_{light}$). The δ values determined for BSE$_{light}$ samples were used in the mass balance model rather than that of the light water added to accommodate for the slight δ offset between these waters. This slight offset may have developed from evaporative fractionation (Allison

et al., 1983) that likely occurred when applying the light water to the recently oven-dried soil within the dry local atmosphere within our lab, or from a small amount of hygroscopic water adsorbed from local atmosphere once soil was removed from the oven (Hillel, 2003). The direction of this slight offset was not consistent with previous observations of isotope effects associated with interactions with clay minerals (Gaj et al., 2017) or carbonates (Meißner et al., 2014). The mass of water remaining in soil samples before high-tension extraction was calculated using gravimetric water contents and the mass of the soil samples after the mid tension centrifuge step. The mass of total water applied to each sample was determined by adding the masses of water remaining in the soil before high tension extraction and water extracted from both centrifuge steps. The mass of light water applied was determined by subtracting the amount of heavy water infused in the sample (covered in section 2.2) from the mass of total water applied.

To assess fractionation associated with evaporation, we calculated the difference in mass for soil filled inserts and corresponding extracted waters. This was evaluated throughout the experiment between centrifuge steps as well as prior to and after equilibration periods. We observed differences of only less than 1% of the mass of the extracted water in all cases, and therefore discounted the impacts of evaporative fractionation on our results and interpretations.

We conducted a pairwise MANOVA between the paired mean $\delta^2H$ and $\delta^{18}O$ isotope values for each of the soil water pools extracted from the three tension ranges, the $\delta$ values of the two applied waters, and the $\delta$ values of waters from $BSE_{light}$ and $BSE_{light+heavy}$ samples. There was a total of seven groups compared against one another at each of the five timepoints.

We further used a time-dependent isotope mixing equation to approximate the time required for soils to completely mix (i.e. reach equilibrium). The model takes the general form:

$$\delta(t) = \delta_e + (\delta_0 - \delta_e)e^{-kt} \tag{3}$$

where $t$ is time since mixing (hour), $\delta(t)$ is the isotope ratio of water extracted at a particular tension by centrifugation or CVD at a particular time point, $\delta_e$ is the equilibrium isotopic ratio expected for the extracted water under perfectly mixed conditions assuming no fractionation or other effects, $\delta_0$ is the isotopic ratio of the extracted sample at time 0, and $k$ is the time or proportionality constant (hour $^{-1}$). Because we were interested in how the isotopic values of waters varied with different tensions, $\delta_0$ and $k$ were allowed to vary based on each extracted water pool (i.e. low tension, mid tension, and high tension). The interaction among the three pools of water in this study within an ecohydrological perspective is diagramed in Fig. 3.

We used data across all experiments to fit Eq. (3), which made initial conditions ($\delta_0$) somewhat uncertain. To account for this error and the expectation that such uncertainties would converge as time went on, we applied a heteroskedastic error term that depends on time since mixing:

$$\sigma = b_0 + \frac{b_1}{t} \tag{4}$$

where $b_0$ and $b_1$ are slope and intercept terms that vary with the different extraction tensions. We determined $\delta_e$ from the mean value of fully mixed water inputs on the left side of Eq. (2) from every two-part mixing model. Mean $\delta$

values and standard deviations used for $\delta_e$ were -57±5‰ for $\delta^2H$ (n=27) and -8.6±0.7‰ for $\delta^{18}O$ (n=27), which are heavily weighted towards the value of the heavy water reflecting the much larger proportion of this water in fully wetted samples.

We compared the distribution of the expected equilibrium value ($\delta_e$) to those of the different extracted fractions to evaluate mixing times. We considered the system to be completely mixed when the median expected $\delta$ value of the different extracted fractions was within the 90th percent credible interval of $\delta_e$.

All statistical analyses were performed with the R v. 3.6.1 software (R Core Team, 2019). The emmeans R package was used to conduct the MANOVA analysis (Lenth, 2019). The time-dependent mixing models were analyzed using the probabilistic programming language Stan (Carpenter et al., 2017), using the rstan programming interface (Stan Development Team, 2019).

## 3 Results

### 3.1 Isotope ratios of extracted waters and MANOVA

The amount of water removed from the soil within each of the tension ranges was consistent across all samples. The low and mid tension centrifuge extractions removed 71±6 % and 17±6 % (n=27) of the soil water, and high tension CVD extraction recovered the remaining 12±1 % (n=27) of the soil water. Average volumes from the three extraction steps in the experiment are illustrated on Fig. 1 in relation to the soil water retention curve for sandy loam soil.

The isotope composition of waters extracted at the three tensions were clearly different at 0 h after soil wetting, but differences diminished with the amount of time the added light and heavy waters were allowed to interact (Fig. 4, Table 1). The isotope ratio of water recovered using CVD of BSE$_{light}$ samples (bulk sample extraction after light water applied) indicates that potentially the water in the sample at this step was altered slightly by evaporative enrichment of heavy isotopes mixed into the oven dried soil, which had a high amount of surface area exposed to dry local atmosphere. Although this changed the isotopic value of water in soil before application of the heavy water, the light waters applied and BSE$_{light}$ extracted waters were not significantly different ($p > 0.05$). The isotope ratio values of the BSE$_{light+heavy}$ samples were not significantly different from that of the heavy waters applied ($p > 0.05$). At 0 h the isotope ratio values of water extracted using centrifugation at low rotational velocity (water extracted at low tension) were not significantly different from those of either the heavy waters applied ($p > 0.05$) or the BSE$_{light+heavy}$ samples ($p > 0.05$). Yet for the samples assessed at 0 h the isotope ratio values among waters extracted across the three different tensions were significantly different from one another ($p$ values < 0.01) (Table 2). After 8 h of mixing the isotope ratio values of water extracted at low tension were significantly different from that of the heavy water applied ($p < 0.05$) and these remained significantly different over the remaining equilibration times ($p$ values < 0.01). After 1 d the isotope ratio values of the waters extracted at low tension were not significantly different from those extracted at mid tension ($p > 0.05$) while the isotope ratio values of water extracted at mid tension were significantly different from those extracted at high tension ($p < 0.01$). After 3 d of mixing the isotope ratio values of waters extracted at low and high tensions remained statistically different ($p = 0.05$), but even these were indistinguishable after 7 d of mixing ($p > 0.05$). Over time the isotopic ratio values for waters recovered from all three tensions converged upon the expected equilibrium value based on mass balance mixing of the two applied waters, predominantly weighted by the

heavy water due to the proportionally much larger amount of heavy water applied. The isotope ratio values of water extracted at high tension were significantly different ($p$ values < 0.01) than BSE$_{light}$ samples, BSE$_{light+heavy}$ samples, heavy water samples and light water samples for all equilibration timepoints. A shortened list of the comparisons between groups is presented in Table 2 and a complete list is found in Appendix A, Table A1.

**3.2 Two-part isotope mass balance model**

The results from the mixing model using Eq. (2) were uniform across soil samples. The mean percent recovered water was 100.2±0.4 % (n=27) based on $\delta^2H$ data with a range of 99.34% to 102.05%, and 100.1±0.1 % (n=27) based on $\delta^{18}O$ data with a range of 99.88% to 100.25%. These values suggest all water applied was accounted for in extraction processes and that minimal, if any, fractionation occurred due to evaporation.

**3.3 Time-dependent mixing model**

Model estimates determined from the time-dependent mixing equation (Eq. (3)) are provided in Fig. 5 and Fig. 6. A 1:1 relationship between observed and predicted values indicates the model did reasonably well at predicting observed values and their uncertainty with only one value observed outside the given uncertainty bound for $\delta^2H$ (Fig. 7). Results were generally consistent between the two isotopes, however $\delta^{18}O$ expressed an upward shift in values as the mixing time proceeded. Mean values of parameters for the time-dependent mixing models are reported in Table 3.

δ²H values at the beginning of the experiment, across tensions, were distinct from one another (Fig. 6). It took about 5 hours for the isotope values of water extracted at low tension to become similar to the expected equilibrium (i.e., well-mixed) $\delta_e$ value. Water extracted at mid tension did not attain a thoroughly mixed value until 12 hours. It took ~104 hours for the water recovered at high tension by CVD to reach the expected equilibration value. These model results suggest it would have taken the sequentially added waters a little more than 4 days to completely mix and equilibrate across the pools of soil water. $\delta^{18}O$ values indicate possible fractionation expressed at day 3 and 7 equilibration timepoints with offsets towards heavier values. Due to these offsets, probability densities were not evaluated with $\delta^{18}O$ data since our time-dependent mixing model did not account for fractionation offsets occurring during equilibration.

**4 Discussion**

Recent work by the ecohydrological community has emphasized the need to understand how the isotopic composition of various pools of water held at a range of tensions interact and evolve over time (Adams et al., 2019; Oerter et al., 2019; Poca et al., 2019). Our approach successfully permitted analysis of the isotopic composition of water extracted at different tensions within a single soil sample offering a method to assess the time-dependent isotopic exchange among soil pools. We believe our approach can be extended to investigate potential isotopic fractionations and chemical exchanges that shape the isotopic and geochemical composition of water in different soil regions over time. Our findings are consistent with those from other recent studies (Adams et al., 2019) suggesting that waters occupying different pore spaces added sequentially to dry soil do not immediately and completely mix. Lags in isotopic mixing and equilibration have implications for studies focused on plant water sources, soil water age or residence times, water

balance, and flux partitioning (Evaristo et al., 2015, 2019; Evaristo and McDonnell, 2019; Good et al., 2015; He et al., 2019; Sprenger et al., 2018; Wang et al., 2019).

The isotopically distinct waters applied to oven-dry soil in our proof-of-concept study required more than 3 days to fully mix and equilibrate. Even with some advection through and out of the soil matrix during centrifugation steps as well as possible minor gravitational downward movement of water during equilibration storage, these results reveal relatively long lag times for complete mixing. Complete mixing would likely take longer for undisturbed soil samples with complex aggregate structure compared to our homogenized and disturbed soil samples. The connectivity

of water pools within and between soil aggregates and other pore regions for undisturbed soil is likely much lower than in disturbed soils where this complex structure has been reduced. The time-dependent mixing model indicated that complete mixing was achieved at ~4.33 days and this timeframe was consistent with the MANOVA results between the waters recovered at the three tensions. However, at 7 days the waters extracted under high tension were significantly different than those of the BSE$_{light+heavy}$ samples (MANOVA), but were within the 90% credible interval

for $\delta^2$H of $\delta_e$ according to the time-dependent mixing model. This highlights a key difference between the statistical methods of comparison: while the MANOVA compares the multivariate normal means across isotopes, our mixing model analysis ignored the $\delta^{18}$O values due to yet unexplained (see below for further discussion) deviations in the mixing model. Nonetheless, while these methods highlight slight differences in their estimate of when the two added waters were completely mixed across all extracted fractions, they both highlight the long time lags in mixing.

The mass balance mixing model revealed that 99% of the water applied to the oven dry soil in our experiment was recovered over the sequence of centrifuge and CVD extractions suggesting minimal losses or isotopic fractionation with evaporation after the soils were completely wetted. We chose to use the isotope value of the bulk water extracted after the light water was applied (BSE$_{light}$) as the end-member in the mass balance model rather than the isotope ratio value of the light water itself. We felt this was justified for the objective of our study, which was to

demonstrate the capability of the combined centrifuge-CVD method to evaluate mixing dynamics among different soil water pools.

    We observed slightly higher $\delta^{18}$O values of the extracted water pools at days 3 and 7 than predicted based on simple mixing of the two waters added to the dry soil (Fig. 5). Because we recovered the expected mass of water (>99%) for these samples, we do not feel the observed $^{18}$O enrichment was a result of evaporation. Water interactions

with clay minerals (Gaj et al., 2017) and carbonates (Meißner et al., 2014), in contrast, typically result in depletion of $^{18}$O in matrix water. The positive shift in $\delta^{18}$O of soil water observed in our study however is consistent with observations reported by Oerter et al. (2014) who found that at low water content $\delta^{18}$O of matrix water increased in the presence of clays enriched with potassium. We cannot discount the possibility of such ionic interactions in our study. The time course for ionic exchanges with clays that influence the oxygen isotope composition of matrix water

might explain why the mixing dynamics observed in our study differed between H and O isotopes. Identifying and analyzing such effects require more thorough analysis.

    Since we limited vapor transport and advection in the current study by holding samples in a closed, isothermal vessel near saturation, we assume the isotope mixing among soil pore waters was dominated primarily by self-diffusion of isotopologues by Brownian motion. This mixing towards equilibrium by self-diffusion in hypothetical

pore space is shown in Fig. 3. Diffusion rate in soil solution is a function of the diffusion coefficient for the solute of interest, a tortuosity factor, volumetric water content ($\theta$) and the solute effective concentration gradient (Chou et al., 2012). We did not measure these variables in our study; rather we simplified the analysis by lumping these processes into a single empirical parameter ($k$) in our time-dependent mixing model (Eq. (3)). However, we expect soil water content as well as other features that determine tortuosity, like aggregate structure and pore size distribution will have

strong influences on the isotopic mixing times of soil water pools. For example, complete mixing in finer textured soils and unsaturated soils will be much longer than those reported here because of these effects, but can be assessed using the general approach we describe.

    Further development of the general approach we present should address potential artifacts related to centrifugation and CVD as a means to extract waters sequentially from a single sample across a range of tensions.

First, the pressure applied to the soil varies within the soil column at a single rotational velocity depending on distance from the center of the centrifuge rotor. This is unavoidable, but potential artifacts may be reduced or avoided by using low-profile centrifuge vessels. Second, the tension by the soil may change between or during centrifugation steps since the proportion of small pores within the soil column increases as pores get compacted to smaller diameters. This also is unavoidable, and the magnitude of this effect on the distribution of isotopically distinct waters recovered at different

tensions should be explored further.

    Additional improvements and expanded applications of the combination approach we present should be considered. For example, use of waters with a greater isotopic difference for experimentally wetting dry soil and reversing the order of the addition of the heavy and light waters would better resolve rates of mixing and possible fractionation effects. Furthermore, applying this combination method to undisturbed soil would need to carefully

consider how soil is sampled before placed in centrifuge inserts. Collecting field samples directly into inserts would minimize compaction and disturbance of aggregate structure. In addition, the oven-drying step could be eliminated, and equilibration could be assessed by using antecedent moisture within undisturbed soil samples. Finally, minimizing the time of centrifugation at each step (Fraters et al., 2017) would provide more highly resolved estimates of soil water mixing times and increase sample throughput. Higher sample throughput is needed since low temporal and spatial

resolution of sampling from the field often limits our ability to thoroughly test mechanisms that create spatial and temporal heterogeneity in the isotopic composition of soil water (Dubbert et al., 2019).

**5 Conclusion**

We present a method for separately extracting water held at different tensions within soil for isotopic analysis and provide a quantitative framework for evaluating time-dependent mixing of isotopically distinct waters within a soil

sample. Our general approach could be extended to provide a means to evaluate the time-dependent interactions among pools of soil water and self-diffusion of water in soils with different soil textures, for undisturbed soil cores that retain complex structure, and under variably saturated conditions. Additional work is needed to refine the application of the centrifuge-CVD combination method for such studies but embracing the general notion of a combination method will overcome perceived limitations unique to each separate extraction technique.


*Code and data availability.* The code and data used in this study can be accessed via Open Science Framework (doi:10.17605/OSF.IO/ET3G5).

*Author contributions.* WHB conducted the experiment, performed data analysis, developed figures, and drafted the paper. JJM helped conceive the experiments, prototyped and refined the centrifugation insert, performed data analysis, and developed the self-diffusion model. MSP provided ideas on experiment design and interpretation of experiment results. DGW helped conceive the experiment and write the paper. All authors edited the paper.

*Competing interests.* The authors declare they have no conflict of interest.

*Acknowledgements.* Support for this work was provided by the National Science Foundation (EPS – 1208909). We thank Dr. Brent Ewers and lab personnel for letting us use their workspace, centrifuge, and rotor. We also thank Dr. Thijs Kelleners for guidance on acquiring soil and use of lab space and equipment. In addition, we would like to thank the University of Wyoming Stable Isotope Facility for assistance with water isotope analysis and access to space and equipment for cryogenic vacuum distillation. Lastly, we would like to thank the Engineering Machine Shop at the University of Wyoming for making the custom inserts.

*Financial support.* This research was supported by the National Science Foundation (EPS – 1208909).

*Review statement.*

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

**Table 1: Mean, standard deviation, and range of isotope values for each extracted water sample from each timepoint**

| Effluent | Timepoint | Mean δ²H ‰ | Mean δ¹⁸O ‰ | Range δ²H ‰ | Range δ¹⁸O ‰ | Number of samples |
|---|---|---|---|---|---|---|
| Low Tension: Centrifuge | 0 hours | $-47 \pm 1$ | $-7.5 \pm 0.3$ | -46 to -50 | -7.0 to -7.9 | 15 |
| | 8 hours | $-53 \pm 1$ | $-7.8 \pm 0.2$ | -52 to -54 | -7.6 to -8 | 3 |
| | 1 day | $-56 \pm 1$ | $-8.0 \pm 0.2$ | -55 to -56 | -7.8 to -8.2 | 3 |
| | 3 days | $-56 \pm 1$ | $-7.8 \pm 0$ | -56 to -57 | -7.8 to -7.8 | 3 |
| | 7 days | $-55 \pm 1$ | $-7.3 \pm 0.3$ | -54 to -56 | -6.9 to -7.5 | 3 |
| Mid Tension: Centrifuge | 0 hours | $-65 \pm 4$ | $-9.2 \pm 0.6$ | -60 to -74 | -8.2 to -10 | 15 |
| | 8 hours | $-63 \pm 5$ | $-8.6 \pm 0.4$ | -58 to -67 | -8.3 to -9 | 3 |
| | 1 day | $-60 \pm 0$ | $-8.3 \pm 0.2$ | -60 to -60 | -8.1 to -8.4 | 3 |
| | 3 days | $-57 \pm 1$ | $-7.9 \pm 0.2$ | -56 to -58 | -7.8 to -8.1 | 3 |
| | 7 days | $-55 \pm 0$ | $-7.0 \pm 0.2$ | -55 to -55 | -6.7 to -7.1 | 3 |
| High Tension: CVD | 0 hours | $-89 \pm 10$ | $-10.8 \pm 1.5$ | -64 to -109 | -6.9 to -13.6 | 15 |
| | 8 hours | $-79 \pm 3$ | $-9.5 \pm 0.4$ | -76 to -82 | -9.0 to -9.7 | 3 |
| | 1 day | $-72 \pm 4$ | $-8.4 \pm 0.2$ | -68 to -75 | -8.2 to -8.6 | 3 |
| | 3 days | $-65 \pm 2$ | $-7.6 \pm 0.6$ | -64 to -67 | -7.0 to -8 | 3 |
| | 7 days | $-62 \pm 2$ | $-6.5 \pm 0.5$ | -61 to -64 | -6.0 to -6.9 | 3 |


**Table 2: The results of pairwise MANOVA tests for the experiment with comparisons between groups of samples, group 1 compared to group 2 on respective rows. Significant values are highlighted in bold, p-value ≤ 0.05. Only showing comparisons that changed from significant to insignificant or vice versa throughout the experiment, while sixteen comparisons not shown stayed either significant or insignificant for all timepoints.**

| MANOVA Comparison | MANOVA Comparison | Timepoint p-values | | | | |
|---|---|---|---|---|---|---|
| Group 1 | Group 2 | 0 hours | 8 hours | 1 day | 3 days | 7 days |
| Mid Tension: Centrifuge | Low Tension: Centrifuge | **<0.0001** | **0.02** | 0.7 | 1 | 1 |
| Mid Tension: Centrifuge | High Tension: CVD | **<0.0001** | **0.0001** | **0.005** | 0.1 | 0.2 |
| Mid Tension: Centrifuge | BSE$_{light+heavy}$ | **<0.0001** | **0.0002** | **0.002** | 0.1 | 0.4 |
| Low Tension: Centrifuge | High Tension: CVD | **<0.0001** | **<0.0001** | **0.0001** | **0.05** | 0.3 |
| Low Tension: Centrifuge | Heavy Water | 1 | **0.05** | **0.001** | **0.001** | **0.004** |



**Table 3: Means and standard deviations of parameters used in the time-dependent mixing model.**

| Isotope | Effluent | $\delta_e$ (‰) | $k$ ($10^2$ hr$^{-1}$) | $b_0$ (10 ‰) | $b_1$ (10 ‰ hr) |
|---|---|---|---|---|---|
| | | | Mean (SD) | | |
| $\delta^2$H | All | -57.4 (4.8) | | | |
| | Low Tension: Centrifuge | | 15.8 (3.2) | 8.9 (2.0) | 5.1 (3.4) |
| | Mid Tension: Centrifuge | | 3.1 (1.0) | 17.7 (5.4) | 29.7 (13.4) |
| | High Tension: CVD | | 1.6 (0.6) | 40.6 (13.9) | 85.7 (36.7) |
| | | | | | |
| $\delta^{18}$O | All | -8.6 (0.7) | | | |
| | Low Tension: Centrifuge | | 0.5 (0.2) | 2.9 (0.6) | 1.1 (10) |
| | Mid Tension: Centrifuge | | 1.5 (0.5) | 2.9 (0.9) | 3.9 (1.8) |
| | High Tension: CVD | | 3.7 (1.4) | 5.7 (1.7) | 10.5 (3.7) |

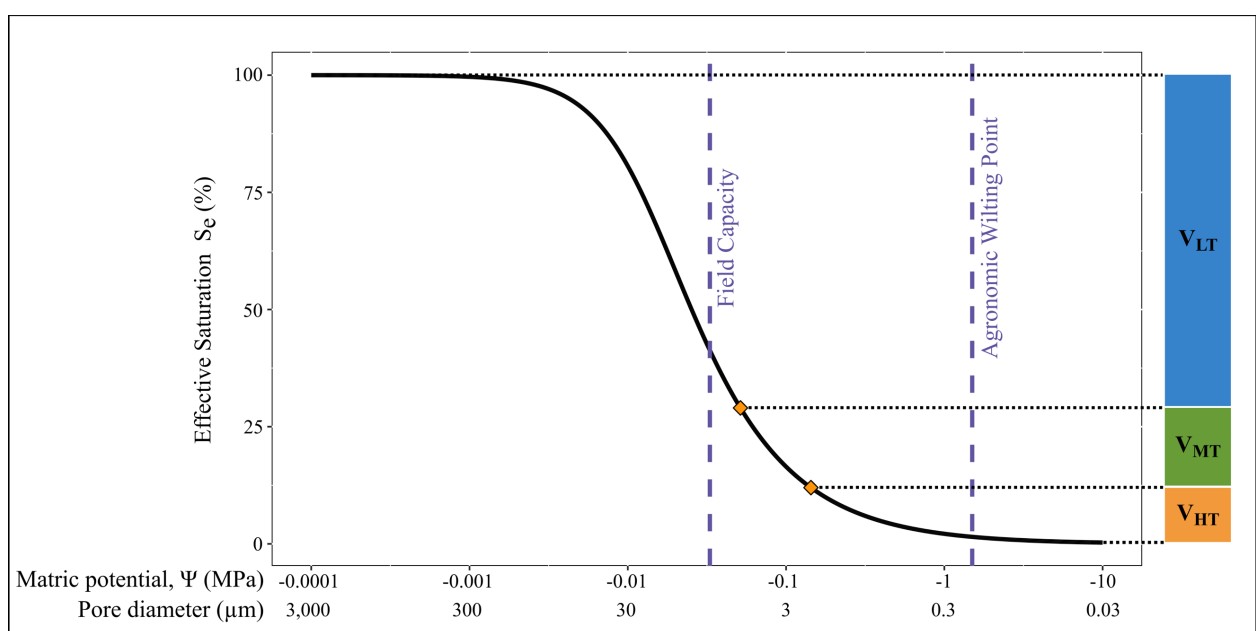

**Figure 1: Soil retention curve for a sandy loam soil using van Genuchten parameters for a general sandy loam (Kosugi et al., 2002). Average volumes (V) from each extraction step of the experiment are illustrated on the right with LT for Low Tension, MT for Mid Tension, and HT for High Tension. Vertical lines are matric potential points of interest: field capacity of -0.033 MPa and agronomic wilting point of -1.5 MPa. The y-axis is effective saturation, a standardized form of volumetric water content. The x-axis has two scales: the top scale is matric potential in MPa and bottom is relative maximum pore size filled at the respective matric potentials (Schjonning, 1992). Samples wetted with both light and heavy waters were near but not at 100% effective saturation.**

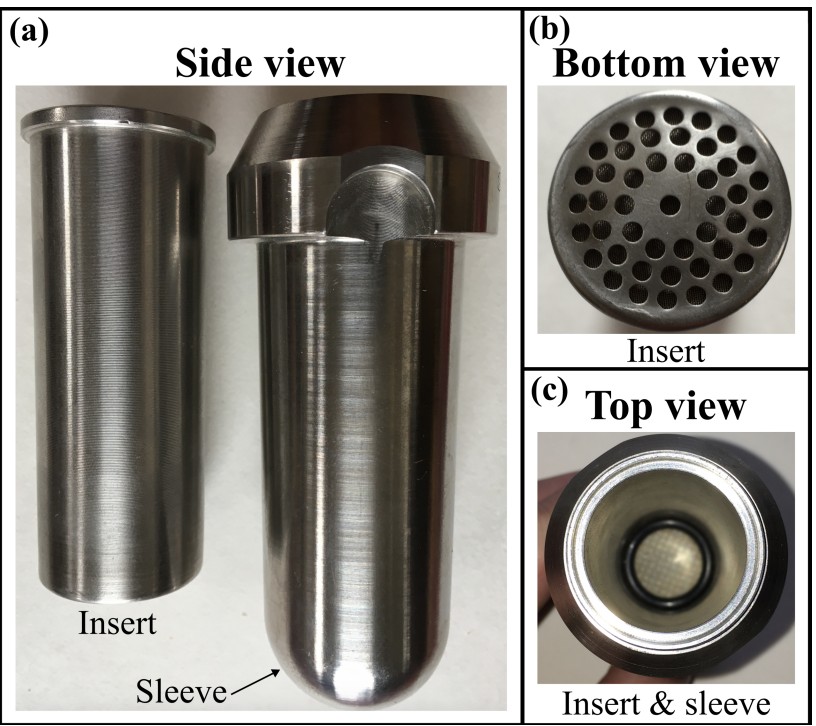

**Figure 2: (a) Image of custom-made centrifuge insert and Sorval sleeve. (b) Bottom view of insert perforated with drilled holes to allow water movement during centrifugation. (c) Top view includes steel screens at bottom of insert secured with rubber o-ring to reduce soil loss during centrifugation. The steel tube inserts were fitted with a collar at the top that secured the position of the insert within the sleeve at roughly 19 mm above the bottom to establish a reservoir for collecting extracted water through the screens and perforated bottom. Small**
**gravity secured caps described in methods section 2.2 collars are not shown in this image.**

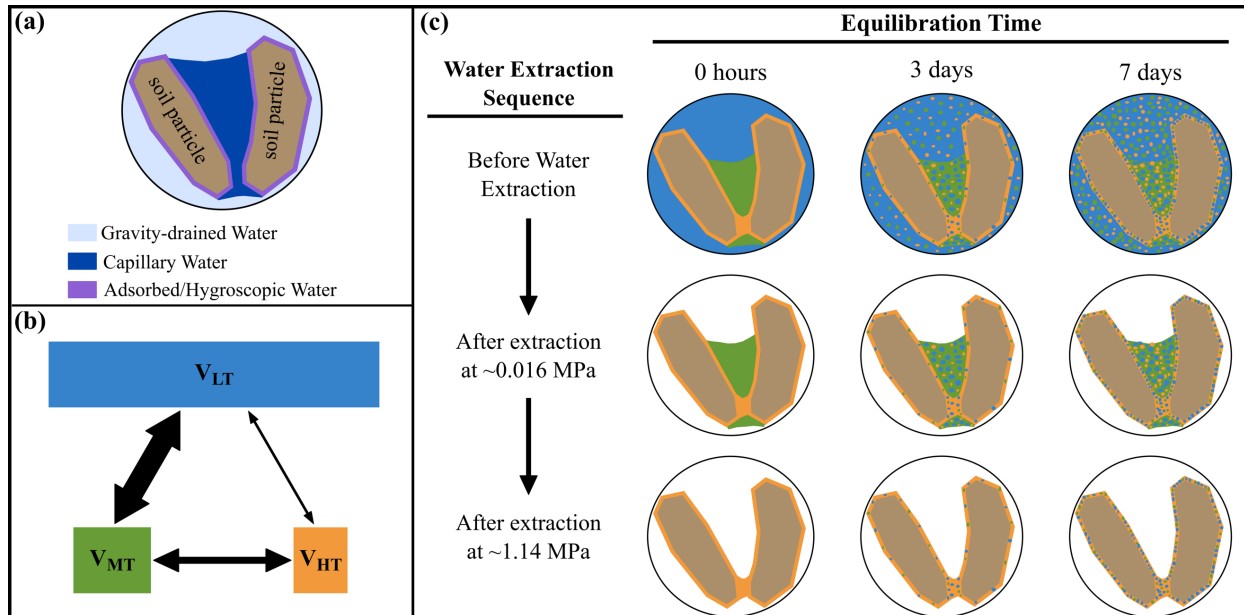

**Figure 3: (a)** Spatial relationship of the three most commonly discussed water pools that make up the bulk water pool in soil near saturation. Absorbed/hygroscopic water, capillary water and gravity-drained water are depicted in hypothetical cross-section view of two soil particles within the soil matrix. **(b)** Relative volumes (V) of soil water pools in this study based on Fig. 1 (LT= low tension, MT= mid tension, and HT= high tension) and the relative amount of interactions (size of black arrows) between pools as equilibration time proceeds. **(c)** Three soil water pools for this study in hypothetical pore space, as diagramed in the first panel, at three equilibration timepoints and various points in the water extraction sequence. Based off of Fig. 1 water extracted at low tension is comprised of gravity-drained water and capillary water, that extracted at mid tension is composed of capillary water, and water extracted at high tension is comprised of capillary water and hygroscopic water. As equilibration time increases, each pool moves closer towards a well-mixed state (i.e. equilibrium).

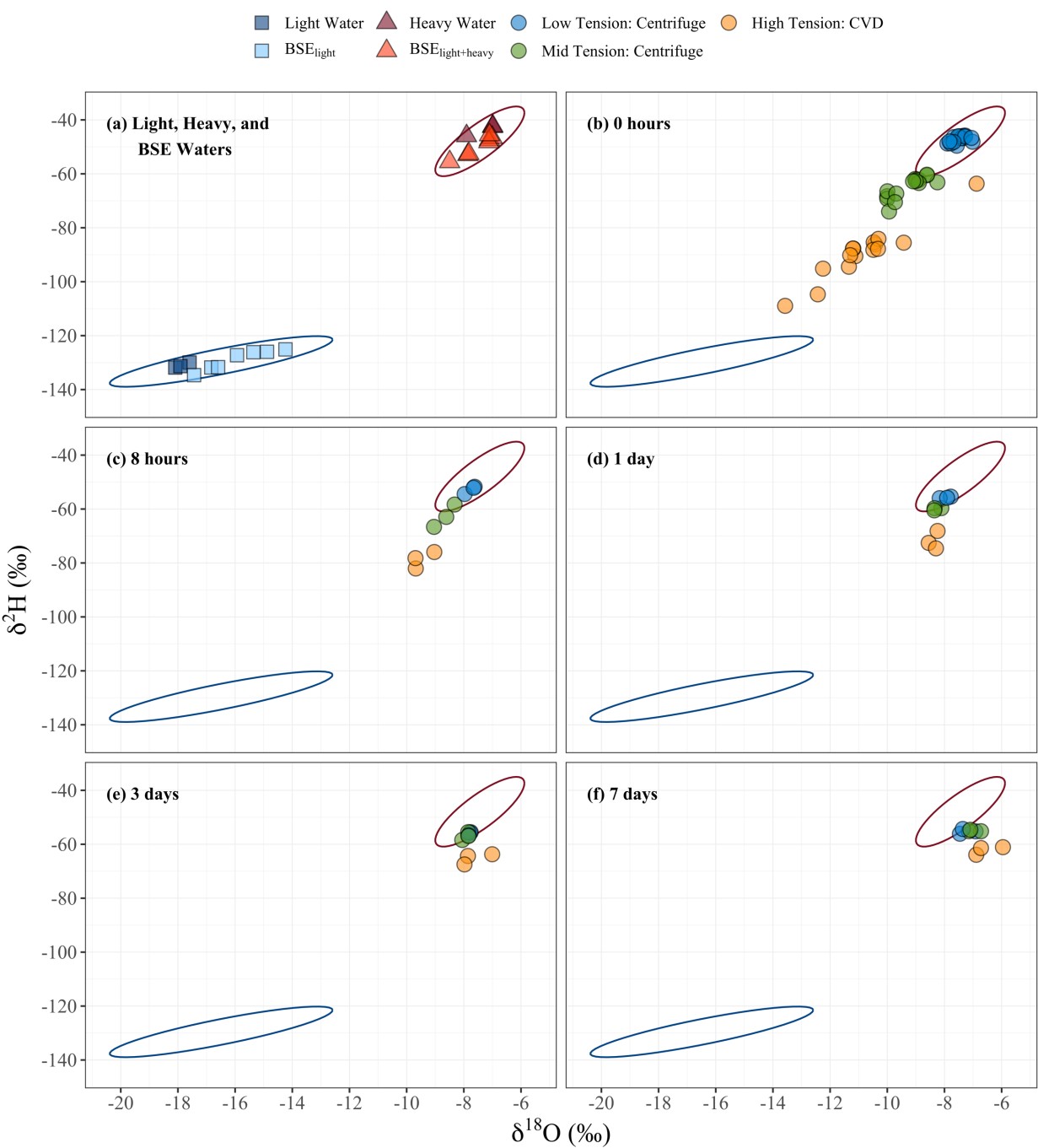


**Figure 4: Isotopic values of water samples in dual-isotope space, δ²H$_{VSMOW}$ (‰) vs. δ¹⁸O$_{VSMOW}$ (‰). (a) Light, Heavy, and BSE(bulk sample extraction) waters with 95% confidence interval ellipses generated by pooled data of Light, Heavy, and BSE waters since the pooled groups were found to be not significantly different with pairwise MANOVA (Table A1 in Appendix) (blue ellipse = BSE$_{light}$ and Light Water, red ellipse = BSE$_{light+heavy}$**
**and Heavy Water). (b-f) Waters extracted at low, mid, and high tension for each equilibration timepoint. The 95% confidence interval ellipses from (a) are included in (b-f) for reference.**

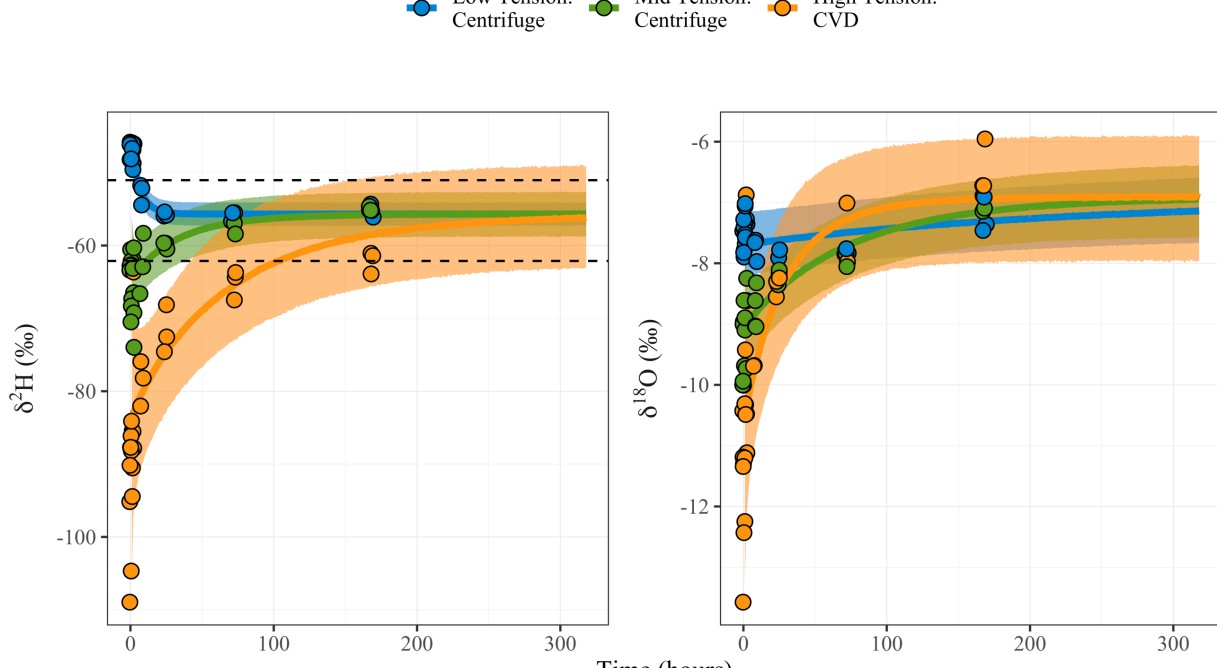

**Figure 5: Time-dependent mixing model curves plotted for $\delta^2H$ and $\delta^{18}O$ (‰, VSMOW) for each extracted water fraction over time. Shaded regions are 90th credible intervals for each curve. The dashed lines are for the 90th credible interval for the equilibrium ($\delta_e$) estimate of $\delta^2H$. $\delta^{18}O$ measured values indicated possible fractionation offset near when equilibrium was achieved according to $\delta^2H$ values. Due to this offset, probability densities with $\delta^{18}O$ data were not evaluated similarly to the $\delta^2H$ values since the time dependent mixing model works under the assumption that there are no fractionation offsets occurring. Therefore, no dashed lines for right plot with $\delta^{18}O$ data.**

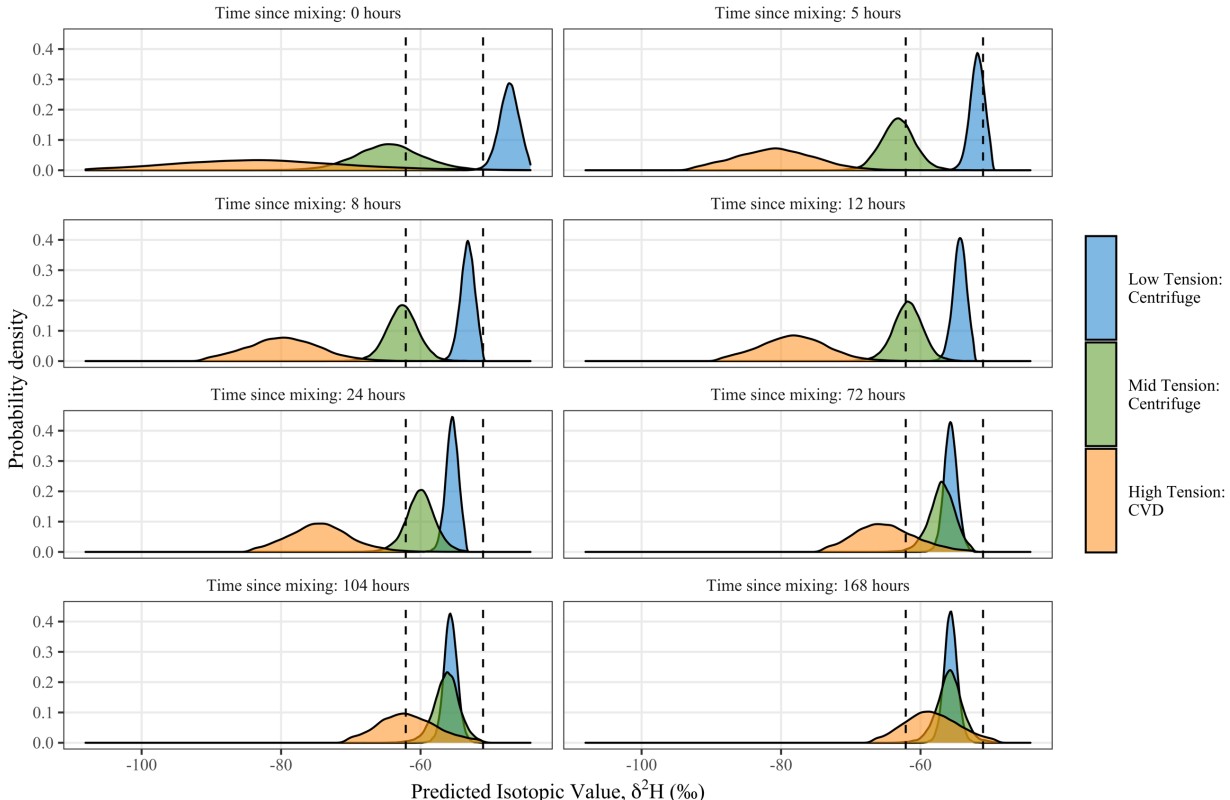

**Figure 6: Time-dependent model for mixing time with distributions of $\delta^2$H for each of the three extracted water fractions over time in relation to the 90th credible interval for equilibrium value ($\delta_e$, dashed lines). Panels include extraction times for the experiment as well as important timepoints for mixing. At 5 hours the median low tension value was within the 90th credible interval of the equilibrium value. At 12 hours, the isotope composition of waters extracted at low and mid tension were similar to the the equilibrium value. It was not until 104 hours (~4.33 days) that the median isotopic value of the water extracted at high tension was also within the 90th credible interval of the equilibrium value.**




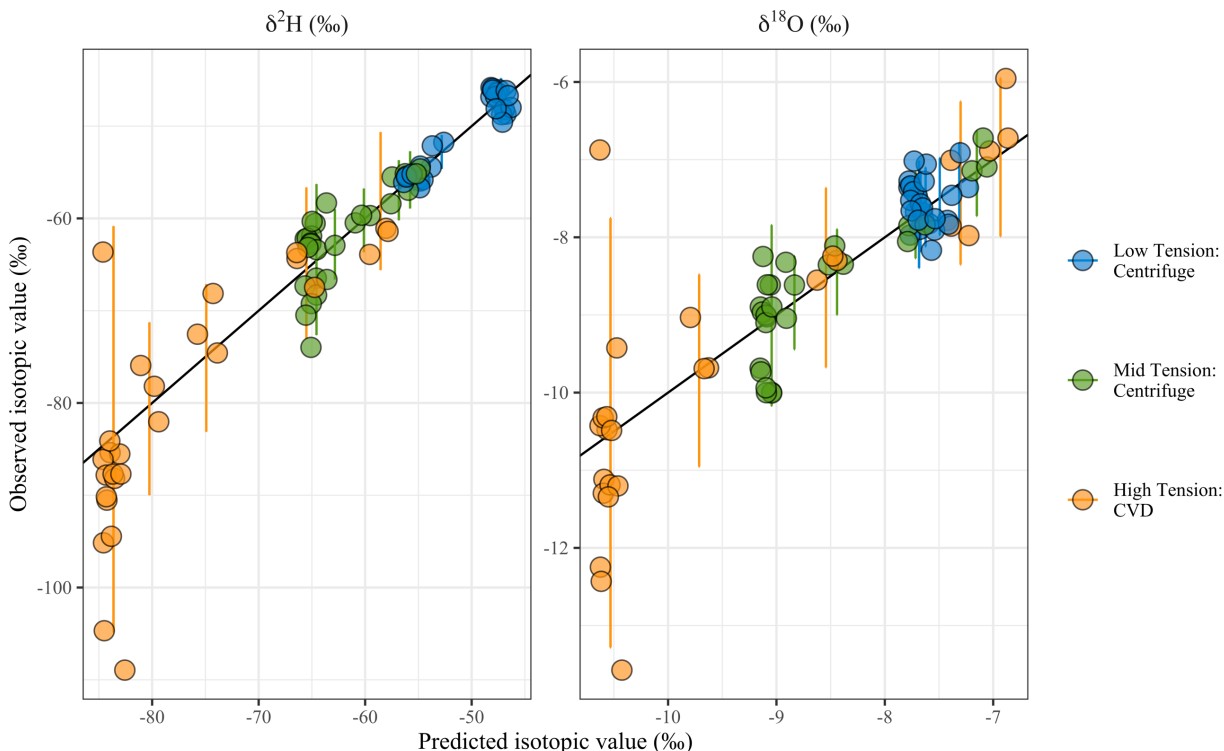

**Figure 7: Comparison of predicted with Eq. (3) and observed values for waters extracted with different tensions. The 1:1 line is shown. Bars represent the credible interval (90%) of the predicted values by timepoint and tension. A slight jitter (3%) has been added to the predicted value (x-axis) in an effort to display the points.**


**Appendix A:**

**Table A1: The p-value results of pairwise MANOVA tests for the experiment with comparisons between groups of samples, group 1 compared to group 2 on respective rows. Significant values are highlighted in bold, p-value ≤ 0.05. Showing comparisons not shown in Table 2.**

| MANOVA Comparison | MANOVA Comparison | Timepoint p-values | | | | |
|---|---|---|---|---|---|---|
| Group 1 | Group 2 | 0 hours | 8 hours | 1 day | 3 days | 7 days |
| Mid Tension: Centrifuge | $BSE_{light}$ | **<0.0001** | **<0.0001** | **<0.0001** | **<0.0001** | **<0.0001** |
| Mid Tension: Centrifuge | Light Water | **<0.0001** | **<0.0001** | **<0.0001** | **<0.0001** | **<0.0001** |
| Mid Tension: Centrifuge | Heavy Water | **<0.0001** | **<0.0001** | **<0.0001** | **0.0004** | **0.006** |
| Low Tension: Centrifuge | $BSE_{light}$ | **<0.0001** | **<0.0001** | **<0.0001** | **<0.0001** | **<0.0001** |
| Low Tension: Centrifuge | $BSE_{light+heavy}$ | 1 | 0.9 | 0.2 | 0.1 | 0.3 |
| Low Tension: Centrifuge | Light Water | **<0.0001** | **<0.0001** | **<0.0001** | **<0.0001** | **<0.0001** |
| High Tension: CVD | $BSE_{light}$ | **<0.0001** | **<0.0001** | **<0.0001** | **<0.0001** | **<0.0001** |
| High Tension: CVD | $BSE_{light+heavy}$ | **<0.0001** | **<0.0001** | **<0.0001** | **<0.0001** | **<0.0001** |
| High Tension: CVD | Light Water | **<0.0001** | **<0.0001** | **<0.0001** | **<0.0001** | **<0.0001** |
| High Tension: CVD | Heavy Water | **<0.0001** | **<0.0001** | **<0.0001** | **<0.0001** | **<0.0001** |
| $BSE_{light}$ | $BSE_{light+heavy}$ | **<0.0001** | **<0.0001** | **<0.0001** | **<0.0001** | **<0.0001** |
| $BSE_{light}$ | Light Water | 1 | 0.9 | 0.8 | 0.8 | 0.8 |
| $BSE_{light}$ | Heavy Water | **<0.0001** | **<0.0001** | **<0.0001** | **<0.0001** | **<0.0001** |
| $BSE_{light+heavy}$ | Light Water | **<0.0001** | **<0.0001** | **<0.0001** | **<0.0001** | **<0.0001** |
| $BSE_{light+heavy}$ | Heavy Water | 0.8 | 0.2 | 0.1 | 0.1 | 0.1 |
| Light Water | Heavy Water | **<0.0001** | **<0.0001** | **<0.0001** | **<0.0001** | **<0.0001** |