# Peer review of "Combination of soil water extraction methods quantifies isotopic mixing of waters held at separate tensions in soil"

_Hydrology and Earth System Sciences, 2019_

## Referee Comment (RC1) · Anonymous Referee #1 · 27 Feb 2020

The manuscript presents the results of an experiment designed to estimate the rate of isotopic mixing in a soil between two waters that differ in their H and O isotope ratios added to soils sequentially following oven-drying. They do show what appears to be a time-dependent process and argue that the time to equilibration is on the order of days (>4 for this soil). I think the manuscript is a contribution to the ongoing and needed effort to better understand the underlying processes that control soil water isotope ratio variation. However, I have what I think are important concerns with the current version.

A key, underlying assumption (that the authors acknowledge) is the absence of fractionation effects associated with water addition after oven-drying or with the extraction

process. While this may be a valid assumption, there is evidence in their results that it's false, particularly for d18O. The authors assess the potential for enrichment as a function of evaporation by mass balance (comparing mass loss with effluent captures) but this does not account for any fractionation effects associated with clay mineral interactions and is itself subject to errors. It is notable that the quantities used in the mass balance calculations were not the isotope ratios of the added waters, but the value of the isotope ratio of the water extracted by CVD immediately after adding the second water. The authors refer to a "slight" offset, but looking at the data in figure 3, there is apparently as much as a 2‰ difference between the "light" water added and the measured CVD-extracted water. This is not a small difference in my view.

I am also curious about the method used to add water. The sequence was: oven dry 350g of soil, add 20 ml of "light" water and mix, subsample into centrifuge inserts and immerse in "heavy" water (presumably completely?). These soils were then presumably saturated. Were they allowed to drain at all before centrifugation, etc.? What is the field capacity of this soil and how does it compare to the amount of water added in the first step? I think it would be useful to know if freely-draining water was part of the pool extracted in the first step.

I think the authors need to more clearly explain their rationale in using the "time-dependent isotope mixing equations." While I see that an exponential fit to the observed data makes sense (at least for d2H) and that there is a tendency for them to converge, I am not sure the logic holds and I think the fitting approach used might obscure the lack of convergence between the CVD data and the centrifuge data (the CVD data plot well below the fitted line in Fig. 4 for d2H on day 7). The idea is that the low –> mid –> CVD represent a gradient from more to less of the recently added "light" water and capture the mixing process as it proceeds. I don't think this approach captures processes that might involve water interacting with clay and I am not convinced that the mixing is "complete" after 4 days based on the results presented in Fig. 4. The authors also acknowledge but do not attempt to explain the very different patterns observed for

d18O. I think there's more to these patterns than it taking longer for H218O and H216O isotopologues to mix than those of H. If this were a simple mixing process, shouldn't both H & O behave similarly in terms of trajectory? I think more careful thought needs to go into interpreting these results. I also think the authors should report the clay mineralogy since multiple authors have suggested potential impacts of clay type on extracted water isotope ratios.

I was surprised to see the "wilting point" value of -1.5 MPa used. I know the authors are aware that this value is quite high (less negative) compared to values many plants adapted to low-water environments can achieve and experience no damage.

I think the authors should clarify what they mean by "precision" and "accuracy" in the isotope analysis section. Presumably the "accuracy" is some measure of how different the measured/corrected values of an internal reference material were relative to a consensus value, but I think this should be explicitly stated. Similarly, the "precision" is presumably some estimate of variance of the reference material (1 standard deviation of how many replicates?), but again this should be stated.

In line 169 the authors refer to "atomic fraction" when I think they mean "isotope ratio" (e.g., 18O/16O).
* * *

---

## Referee Comment (RC2) · Anonymous Referee #2 · 3 Mar 2020

This paper presents some progress on the centrifuge technique to separate soil water held at different bindings strengths into the potential stable isotopic pools that may exist in soils. I think this study has some good contributions to offer, but I also think it needs some improvement before I can endorse its publication in HESS.

General comments:

I found the analysis and discussion of the results to be quite "thin". By that, I mean that there is not an especially in depth or nuanced explanation and discussion of many components throughout. Specific examples follow, but in general, I suggest that the senior authors of the manuscript return to it with a more discriminating eye and identify

where it can be "deepened".

The authors base the rational for conducting the study on making progress on identifying the potential soil water reservoirs (isotopic or otherwise) that underpin the ecohydrologic separation, or "Two Water Worlds" (TWW) hypothesis. However, there is only the most minor discussion of this concept in the introduction, and then the authors return to it throughout the results and discussion citing how their findings apply to TWW. This is problematic because the reader doesn't have any firm understating of TWW or how the authors are interpreting TWW (because interps vary). I suggest there be a fuller discussion of TWW and how this study specifically contributes to investigating it in the introduction.

Because HESS has an open review process, subsequent reviewers have the advantage of seeing previous reviewer's comments. That is the case here, and while I do not intend to "pile on" the authors, I do support Reviewer #1's comments, especially in regards to the mixing analysis (see my specific comments below).

Specific Comments:

L44: Need a brief explanation of what in situ equilibration is and some references of papers using either of these methods.

L100: These waters aren't all that different in isotope compositions. Nota Bene: Kona Deep drinking water is about 0 ‰ in $\delta$18O and $\delta$2H and is available on Amazon.

L107: The abstract claims that the light water was enough volume to fill only the smallest pores. The procedure described here seems very arbitrary.

How do you have any confidence or measure of what soil pores where filled and to what extent?

L118: Are you sure this was the extraction temp? Did you use boiling water? Laramie is pretty high elevation and thus water has a low boiling point.

L122: 95% is still not ALL of the water.

L127: You use the Two Water Worlds terminology here, but you haven't ever really discussed t in any detail in the introduction. I suggest you do so, to help contextualize the rest of the paper.

L136: Three and four hours seems like a long time! On what basis did you choose these times?

L138: You never really discuss what is tightly or highly bound, or what the potential mechanisms for this soil water are. There are many aspects to this, from soil pore size, to soil mineralogy, etc. This is a main concept of your paper, but you never give the reader any background or basis of understanding how you are using this terminology and "boundness" concepts.

L140: Again, are you sure it was ALL of the water left. Or was it 95%? I dont mean to be tedious here, and there are limits to CVD, but that is prwcisely my point. Even CVD at 100 C wont get all the water out that is in interlayers spaces in clays, etc. Are more nuanced discussion is needed (maybe it comes later in the discussion), and at least some acknowledgement of the study's potential limitations is needed. I will look for that as I read. . .

L142: I think you should move up the details about the centrifuge and inserts. It hard to envision what you did until you tell us about the inserts.

L149: Good that you accounted for evap during the procedure. I assume it was done at room temp, but I could easily see temp being higher inside the centrifuge, especially for 3 to 4 hours. Did you measure this?

L167: What is atomic fraction? Do you mean isotope ratios (not in delta format)? Or do you mean mixing fraction?

L178: Is this 1% the total mass (water + soil) or just water? If it was total mass, then a decent amount of water lost to evap (and a big shift in isotope ratios) could be contained

in the 1% number. I suggest a sensitivity analysis be done to quantify (in isotope terms) what the effects of this much water loss would actually be. It may seem tedious and unnecessary, but with this much handllng of the wet soil, I could easily see evaporation being a bigger factor in isotope results than a casual view would expect.

L217: You state that the BSE waters were not significantly different from the applied waters, but in Figure 3 upper left panel they sure look different to me. It seems that you were not getting back what you put in. This seems problematic.

L218 / Figure 3: Suggest adding A -F labels to the panels in Figure 3. Also, the x-axis labels and ticks seem inadequate.

L222: I suggest keeping the applied water points in all panels in Figure 3 for easier comparison. Perhaps make them dashed outline or ghosted or something to show them but not distract form the time series data.

L227: So basically, after enough time, all the waters extracted by any means all converged upon the Heavy water signature. And the heavy water signature is the one that you soaked the sample in, but only put a little of the light water in the same samples?

L231: So, the conclusion is that the samples were all well mixed? OR something else? Because the dont look well mixed to me, especially not until dy 3 or later. Am I missing the point? If so, please explain better.

L243: How do you evaluate the mixing results if you dont actually known how much of each type of water you put into the soil? Seems to me that with so much more heavy water than light, you are not really evaluating mixing, but more like the time to equilibrium, wherein the heavy water signal just overwhelmed the light because there was so much more of it.

L248: This section reads more like a conclusions paragraph than the start of a discussion. You haven't really supported any of these statements, yet.

L252: What are the proposed mechanisms of mixing? This is hard to determine, because you haven't ever discussed where in soil water is actually held. Is the "mixing" done via diffusion? if so, water self-diffusion in soils is fairly well studied and you could greatly increase the impact of your findings by bringing in some discussion of that work. This seems like an over simplistic analysis of your results, which are a bit fast and loose as it is. No offense intended, just that I am seeking more detail and justification in your measurements and results.

L266: Are there carbonates in your soil? Easy test with HCl.

L270: This is the first time you have acknowledged that your samples have perturbed soil structure and thus pore sizes. This may be the biggest reason for any isotope effect of any discussed.

L277: Finally, the discussion I was yearning for. Can you expand by making some calculations that support these arm waving statements?

L288: Good point on the pore size changing during the spinning.

L290: Yes, shorter spin times!
* * *

---

## Author Comment (AC1) · 23 Apr 2020

April 22nd, 2020

Dear Dr. Josie Geris,

Please find our author comments for the manuscript Combination of soil water extraction methods quantifies isotopic mixing of water held at separate tensions in soil. The comments from both anonymous referees were very insightful and have helped highlight how to improve the manuscript.

Following the guidelines, we have responded to each referee comment and when nec-

essary we have indicated our planned changes to the manuscript. Given the reviewers comments, we intend to: • Add more information related to the two water worlds (TWW) hypothesis to the introduction, as well as provide additional insights, relevant to our research, in the discussion; • Provide a more complete discussion of the soil physical processes important in our analysis – again adding needed information in the introduction and discussion sections; • Highlight in the discussion additional issues related to fractionation that are still not completely resolved (e.g., missing processes not included in our time-dependent self-diffusion model) We hope the planned changes will help future readers understand how this method relates to helping the community address the TWW hypothesis as well as the limitations and future directions that should be considered by the community.

Sincerely,

William Bowers, Jason Mercer, Mark Pleasants, and David Williams

Thank you for your positive comments on our manuscript and we hope that we will have a chance to revise the manuscript as we think we can address all the comments raised by both Referee #1 and Referee #2.

Author's responses to anonymous referee #1:

(1) The manuscript presents the results of an experiment designed to estimate the rate of isotopic mixing in a soil between two waters that differ in their H and O isotope ratios added to soils sequentially following oven-drying. They do show what appears to be a time-dependent process and argue that the time to equilibration is on the order of days (>4 for this soil). I think the manuscript is a contribution to the ongoing and needed effort to better understand the underlying processes that control soil water isotope ratio variation. However, I have what I think are important concerns with the current version.

A key, underlying assumption (that the authors acknowledge) is the absence of fractionation effects associated with water addition after oven-drying or with the extraction

process. While this may be a valid assumption, there is evidence in their results that it's false, particularly for d18O. The authors assess the potential for enrichment as a function of evaporation by mass balance (comparing mass loss with effluent captures) but this does not account for any fractionation effects associated with clay mineral interactions and is itself subject to errors. It is notable that the quantities used in the mass balance calculations were not the isotope ratios of the added waters, but the value of the isotope ratio of the water extracted by CVD immediately after adding the second water. The authors refer to a "slight" offset, but looking at the data in figure 3, there is apparently as much as a 2‰ difference between the "light" water added and the measured CVD-extracted water. This is not a small difference in my view.

Response: We appreciate your insight and acknowledge the potential fractionations that may be expressed as water interacts with clay particles. However, based on observations reported in the literature such effects would likely cause fractionations to occur in a different direction than the slight offset we report here (Gaj et al., 2017). The offset between "light" water applied soil and water extracted from soil with "light" water (BSElight) was in the direction that is typical of fractionation due to evaporation (Allison et al., 1983). This evaporation is likely due to the small amount of water applied to exposed soil samples in the dry atmosphere of our laboratory environment in Laramie, WY. Our soil had roughly 9% clay, and water extracted from clay-rich soils are generally observed to be depleted in heavy isotopes (Gaj et al., 2017), not enriched as we observed. Thus, we feel the offset observed at this stage of our experimental water additions was more likely due to slight evaporation, and we therefore feel justified to exclude this effect from the mass balance determinations. Since the water had changed likely due to evaporation after application, we felt it was more realistic to use BSElight water as input in two-part mass balance determinations. More details on observed offset in d18O for samples with longer equilibration periods is addressed in response to referee #1 general comment 3.

Planned changes: A statement will be added in the methods section 2.5 after acknowledging the slight offset that states "This slight offset could be explained by evaporative fractionation of both hydrogen and oxygen isotopes that likely occurred when applying the "light" water to the recently oven dried soil and is inconsistent with expected isotope effects due to interactions with clay minerals (Gaj et al., 2017), or carbonates (Meißner et al., 2014) within the study soil." Further explanation of the fractionation effects considered in this study will be included in a new paragraph of the discussion which will go over this slight offset between "light" water and BSElight extracted water as well as the offsets in d18O data that were observed for samples with longer equilibration periods (discussed more in response to referee #1 general comment 3).

(2) I am also curious about the method used to add water. The sequence was: oven dry 350g of soil, add 20 ml of "light" water and mix, subsample into centrifuge inserts and immerse in "heavy" water (presumably completely?). These soils were then presumably saturated. Were they allowed to drain at all before centrifugation, etc.? What is the field capacity of this soil and how does it compare to the amount of water added in the first step? I think it would be useful to know if freely-draining water was part of the pool extracted in the first step.

Response: Thanks for raising your concerns and we agree that more details on the procedure will benefit readers and future application of the procedure. In short, the pool extracted in the first step does contain some freely-draining water, or water above field capacity. Centrifuge inserts were not immersed completely; they were placed in an open container and "heavy" water was poured into the container to a level just below the top of the soil level in the centrifuge inserts. Once the top soil layer of each insert was visibly fully wetted, the inserts were removed from the container. This process allowed the inserts to be filled from the bottom up via the drilled holes on bottom of inserts to reduce the amount of air that could be caught in soil pores. Inserts were not allowed to fully drain before centrifuge steps. According to figure 2, field capacity would drain more than 50% of the effective saturation for a sandy loam soil and with this in mind the light water added would be much less than the field capacity of the soil.

Inserts were weighed carefully on the scale and placed either into the centrifuge quickly or into the airtight container for storage until being put through the centrifuge at a later time point. There was inherently a small amount of drainage from the holes at bottom of inserts that occurred when removing inserts from container with "heavy" water during the weighing process and during storage. The difference in weight of this drained water was not considered in total weight of samples. We also see a need to clarify that water samples were near saturation and not at saturation as it would likely take longer for soils to reach complete saturation and due to minor drainage from inserts after fully wetting soil. Minor drainage happened when removing inserts from container with "heavy" water due to cohesive forces between water molecules and uncovered holes at bottom of inserts. Minor drainage also happened on the scale during weighing before centrifugation or during storage and was very minimal for both. All minor drainages combined are estimated to be roughly 3-5% of total water weight and would not have a significant impact on our results. Also, the weight of water drained onto scale was not considered in the weight of insert recorded before centrifugation for all mass balance calculations.

Planned changes: More details on the wetting process with "heavy" water will be included in the methods section 2.2 as well as how the minor drainage during removal from container used to fully wet samples with "heavy" water, minor drainage during weighing before centrifugation, and minor drainage during storage were not considered in total weight. We will include how the weight of water drained onto scale was not considered in the weight of insert recorded before centrifugation for all mass balance calculations. Due to these minor drainages we will change the verbiage used throughout the manuscript to say that samples were fully wetted and near saturation rather than at saturation before extraction via centrifugation or sample storage for equilibration time. To maintain consistency within manuscript, caption of Fig. 2 will include that for our study the samples were near 100% effective saturation but not at 100%. We will explicitly state that the low tension fraction includes freely draining water and tie this together with definitions used for mobile and matrix water. These details will

also be addressed in the discussion since the gravitational downward movement of some water may have influenced mixing as well and will help improve the method moving forward. In addition, we will encourage future studies to place saturated inserts into sleeves immediately after fully wetting to capture more of the freely draining water, but highlight that this was not possible for our study design because we would have needed 8 sleeves and only had 4 sleeves. To capture all freely draining water or truly get saturated conditions before extraction, holes at bottom of inserts would need to be covered before removing from container with "heavy" water or a different method for wetting samples would need to be developed.

(3) I think the authors need to more clearly explain their rationale in using the "time-dependent isotope mixing equations." While I see that an exponential fit to the observed data makes sense (at least for d2H) and that there is a tendency for them to converge, I am not sure the logic holds and I think the fitting approach used might obscure the lack of convergence between the CVD data and the centrifuge data (the CVD data plot well below the fitted line in Fig. 4 for d2H on day 7). The idea is that the low –> mid –> CVD represent a gradient from more to less of the recently added "light" water and capture the mixing process as it proceeds. I don't think this approach captures processes that might involve water interacting with clay and I am not convinced that the mixing is "complete" after 4 days based on the results presented in Fig. 4. The authors also acknowledge but do not attempt to explain the very different patterns observed for d18O. I think there's more to these patterns than it taking longer for H218O and H216O isotopologues to mix than those of H. If this were a simple mixing process, shouldn't both H & O behave similarly in terms of trajectory? I think more careful thought needs to go into interpreting these results. I also think the authors should report the clay mineralogy since multiple authors have suggested potential impacts of clay type on extracted water isotope ratios.

Response: We appreciate these concerns. To be clear, the idea is that the low –> mid –> CVD represents a gradient from less to more of the first added "light" water

and captures the mixing process as it proceeds. Time dependent isotope mixing equations were used because there was a clear change in isotope signature of low-, mid-and high-tension effluents over time and we wanted to see if the process could be explained simply for this proof of concept study. The fitting approach takes into account that the initial conditions were inherently different for each time the experiment was done and because of this there are uncertainty bands around each modeled line (refer back to equation 4 with the heteroskedastic error term). We agree the simple model suggests that other processes need to be accounted for in the equilibration/mixing process and we need to incorporate more nuances in the discussion of our manuscript. As mentioned in the discussion (line 264), we cannot explain the offset in d18O data with previous observations of isotope effects due to interactions with clay minerals (Gaj et al., 2017) or carbonates (Meißner et al., 2014) as those have been observed to deplete extracted waters in d18O. However, after further exploration of isotope fractionation effects from interactions with ions, the shift in our d18O could be due to interactions with K+ or similar cations with lower ionic potential. But, the shift is greater than previous observations by Oerter et al., 2014, which did not exceed 1 per mil enrichment like that of our data.

Planned changes: We will modify the current statements in the discussion (currently around line 264) to not include that isotope effects are not explained by interactions with ions. This will be done by expanding the discussion to have a separate paragraph that covers the nuances of the isotope effects observed and possible explanations. We will explain that they do shift in similar direction to previous observations related of interactions with K+ (Oerter et al., 2014), but are of greater magnitude than the enrichment previously observed. This greater magnitude could be due to longer equilibration time before extraction and possibly due to ions being dissolved into solution over time within the soil. We will urge the community to consider the time of equilibration used when assessing isotope effects due to interactions with minerals and chemical constituents in soil. We will also provide a suggestion for future studies to consider using more inert soil or soil with a high concentration of K+ for future applications of the method that aim

to evaluate this in more detail. The offset is present in all tension ranges of extractions (low, mid, and high), but is more evident in the high-tension fraction, so it is not solely due to CVD extractions. In addition, this extra paragraph will discuss how the simple time-dependent mixing model suggests that other processes need to be accounted for within future models. One suggestion would be to incorporate the mass ratio with $\partial(t)$ and $\partial 0$ as the amount of isotopologues in each tension range effluent (low, mid, and high) may drive the rate of diffusion. Another suggestion would be to incorporate terms from the equation for diffusion of solutes in soil (covered in more detail in response to referee #2 specific comment L277). To help frame this discussion, we hope to also include a new figure of a conceptual box model for each of the tension ranges and how we interpret the extracted fractions (each as a box) interacting over time and the amount of mixing that occurs between each fraction (box) in relation to equation for rate of diffusion of solutes in soil and soil physical properties, like pore size. However, we will also go over the nuances of how this is commonly depicted in a simplified manner, especially in the case of two water worlds hypothesis, as separate fractions when it really should be considered a continuum (Sprenger et al., 2018).

(4) I was surprised to see the "wilting point" value of -1.5 MPa used. I know the authors are aware that this value is quite high (less negative) compared to values many plants adapted to low-water environments can achieve and experience no damage.

Response: We agree that this pressure is not the wilting point for all plants under all soil conditions. We did state in the methods that wilting point and field capacity vary from soil-to-soil and plant-to-plant (line 125).

Planned changes: We will clarify that this is considered the standardized reference value of wilting point used mostly in the agronomic literature and not technically the wilting point for all plants. In addition, as a response to specific comment by referee #2 of line 138 as well as their first general comment, we will add definitions for 'matrix' and 'mobile' water. These definitions from the recent literature of Brantley et al., 2017 use the agronomic wilting point of -1.5 MPa as a reference point.

(5) I think the authors should clarify what they mean by "precision" and "accuracy" in the isotope analysis section. Presumably the "accuracy" is some measure of how different the measured/corrected values of an internal reference material were relative to a consensus value, but I think this should be explicitly stated. Similarly, the "precision" is presumably some estimate of variance of the reference material (1 standard deviation of how many replicates?), but again this should be stated.

Response: We appreciate this comment and agree that we were not clear about how we expressed accuracy and precision of the isotope ratio measurements. Indeed, accuracy is a measure of how much the mean measured/corrected values of d2H and d18O of the reference water deviate from the known calibrated values of that reference water, while precision is the standard deviation for all measured values of the lab reference water acquired while analyzing the unknown samples.

Planned changes: We will provide additional clarification as "We report the accuracy as the absolute difference between the mean of analyzed lab reference water samples (n=15) and the calibrated value of lab reference water. We reported precision as the standard deviation of all lab reference water samples analyzed (n=15)."

(6) In line 169 the authors refer to "atomic fraction" when I think they mean "isotope ratio" (e.g., 18O/16O).

Response: Thank you for this comment as this was an error that needed to be addressed since this should be fractional abundance. We used the reference Hayes 2004, which highlights this with equations 5 and 5a on fractional abundance as well as how fractional abundance was and can be used for mass balance equations similar to equation 7 in Hayes 2004.

Planned changes: To help reduce potential confusion, we will change this to isotope ratio because we get the same results when performing two-part mass balance mixing model with isotope ratio as with fractional abundance.

References:

Allison, G.B., Barnes, C.J., Hughes, M.W., 1983. The distribution of deuterium and 18O in dry soils 2. Experimental. J. Hydrol.

Brantley, S.L., Eissenstat, D.M., Marshall, J.A., Evaristo, J., Balogh-Brunstad, Z., Dawson, T.E., McDonnell, J.J., Godsey, S.E., Karwan, D.L., Weathers, K.C., Chadwick, O., Roering, J., Papuga, S.A., 2017. Reviews and syntheses: on the roles trees play in building and plumbing the critical zone. Biogeosciences 14, 5115–5142.

Gaj, M., Kaufhold, S., Koeniger, P., Beyer, M., Weiler, M., Himmelsbach, T., 2017. Mineral mediated isotope fractionation of soil water. Rapid Commun. Mass Spectrom. 31, 269–280.

Hayes, J.M., 2004. An Introduction to Isotopic Calculations. Woods Hole Oceanogr. Institution, Woods Hole , MA 2543, 1–10.

Meißner, M., Köhler, M., Schwendenmann, L., Hölscher, D., Dyckmans, J., 2014. Soil water uptake by trees using water stable isotopes ($\delta$2H and $\delta$18O)$-$a method test regarding soil moisture, texture and carbonate. Plant Soil 376, 327–335.

Oerter, E., Finstad, K., Schaefer, J., Goldsmith, G.R., Dawson, T., Amundson, R., 2014. Oxygen isotope fractionation effects in soil water via interaction with cations (Mg, Ca, K, Na) adsorbed to phyllosilicate clay minerals. J. Hydrol. 515, 1–9.

Sprenger, M., Tetzlaff, D., Buttle, J., Laudon, H., Leistert, H., Mitchell, C.P.J., Snelgrove, J., Weiler, M., Soulsby, C., 2018. Measuring and Modeling Stable Isotopes of Mobile and Bulk Soil Water. Vadose Zo. J. 17, 0.

---

## Author Comment (AC2) · 23 Apr 2020

April 22nd, 2020

Dear Dr. Josie Geris,

Please find our author comments for the manuscript Combination of soil water extraction methods quantifies isotopic mixing of water held at separate tensions in soil. The comments from both anonymous referees were very insightful and have helped highlight how to improve the manuscript.

Following the guidelines, we have responded to each referee comment and when nec-

essary we have indicated our planned changes to the manuscript. Given the reviewers comments, we intend to: • Add more information related to the two water worlds (TWW) hypothesis to the introduction, as well as provide additional insights, relevant to our research, in the discussion; • Provide a more complete discussion of the soil physical processes important in our analysis – again adding needed information in the introduction and discussion sections; • Highlight in the discussion additional issues related to fractionation that are still not completely resolved (e.g., missing processes not included in our time-dependent self-diffusion model) We hope the planned changes will help future readers understand how this method relates to helping the community address the TWW hypothesis as well as the limitations and future directions that should be considered by the community.

Sincerely,

William Bowers, Jason Mercer, Mark Pleasants, and David Williams

Thank you for your positive comments on our manuscript and we hope that we will have a chance to revise the manuscript as we think we can address all the comments raised by both Referee #1 and Referee #2.

Author's responses to anonymous referee #2:

This paper presents some progress on the centrifuge technique to separate soil water held at different bindings strengths into the potential stable isotopic pools that may exist in soils. I think this study has some good contributions to offer, but I also think it needs some improvement before I can endorse its publication in HESS.

General comments:

(1) I found the analysis and discussion of the results to be quite "thin". By that, I mean that there is not an especially in depth or nuanced explanation and discussion of many components throughout. Specific examples follow, but in general, I suggest that the senior authors of the manuscript return to it with a more discriminating eye and identify

where it can be "deepened".

The authors base the rational for conducting the study on making progress on identifying the potential soil water reservoirs (isotopic or otherwise) that underpin the Ecohydrologic separation, or "Two Water Worlds" (TWW) hypothesis. However, there is only the most minor discussion of this concept in the introduction, and then the authors return to it throughout the results and discussion citing how their findings apply to TWW. This is problematic because the reader doesn't have any firm understating of TWW or how the authors are interpreting TWW (because interps vary). I suggest there be a fuller discussion of TWW and how this study specifically contributes to investigating it in the introduction. Because HESS has an open review process, subsequent reviewers have the advantage of seeing previous reviewer's comments. That is the case here, and while I do not intend to "pile on" the authors, I do support Reviewer #1's comments, especially in regards to the mixing analysis (see my specific comments below).

Response: We very much appreciate this comment as we think that it is important to clearly define the benefits this method may have for studies exploring the mechanisms causing ecohydrologic separation, beyond simply reporting its existence.

Planned changes: The introduction will be revised to include more background on the role of soil physics when discussing mechanisms influencing soil water transport that would generate observed patterns consistent with the Two Water Worlds hypothesis and the interpretation that we are trying to address. To clarify, we interpret the Two Water Worlds hypothesis similar to that in lines 33-36 where we refer to the original introduction of the hypothesis by Brooks et al., 2010 and include updated verbiage by Brantley et al., 2017 that plants are accessing matrix water that is incompletely mixed with isotopically distinct mobile soil water. This interpretation will be highlighted more as we will include more detailed definitions of matrix water and mobile water in reference to the ranges of soil matric potentials at which these "pools" are defined by the literature we cited (please see response to referee #2 specific comment L138 for more on these definitions). In addition, we will explicitly state that the observations mentioned

in lines 33-36 support the Two Water Worlds hypothesis. We will also provide context to Two Water Worlds/ecohydrologic separation studies which have relied on methods discussed in introduction (particularly near lines 57-58). In addition, there will be a new paragraph in the discussion that highlights what our findings mean for interpreting the TWW hypothesis. Please also see response to referee #1 general comment 3, where we mention a new conceptual model diagram that will be referenced when discussing the mechanisms of water mixing between different sized pores and tension fractions in relation to our study and two water worlds hypothesis.

Specific Comments:

L44: Need a brief explanation of what in situ equilibration is and some references of papers using either of these methods.

Response: Thanks for your suggestion and we agree that adding a reference and more details here will help the reader of this article.

Planned changes: We will refer to (Oerter and Bowen, 2017) in a statement highlighting how in situ direct equilibration utilizes field-based water vapor laser spectroscopy with the assumption that most mobile soil water is in isotopic equilibrium with soil water vapor.

L100: These waters aren't all that different in isotope compositions. Nota Bene: Kona Deep drinking water is about 0 ‰ in _18O and _2H and is available on Amazon.

Response: We appreciate this comment and agree that a greater isotopic difference between waters could be a fascinating avenue for future studies exploring the processes of water mixing/self-diffusion in soil. For the purposes of this study the differences between "heavy" and "light" waters provided us with initial conditions at "0 hours" where the effluents from each tension range (low, mid, and high) did not overlap. In addition, these differences were closer to what might be expected in natural field settings in environments that receive seasonal precipitation as snow in winter and rain in

summer.

Planned changes: We will add a short statement, when introducing the waters used in the experiment within methods section, that these waters where chosen due to the differences and relative isotopic values being similar to isotopic values and differences for precipitation input expected in temperate environments that receive snow in winter and rain in summer. Within the discussion, we will suggest future studies to consider greater isotopic difference in applied waters as the difference in proportion of isotopologues may dictate the rate of self-diffusion. In addition, our planned addition to discussion about time-dependent mixing model (covered in more detail in response to referee #1 general comment 3) will include how our model does not include mass differences between fractions of extracted waters and the degree to which they interact and equilibrate over time.

L107: The abstract claims that the light water was enough volume to fill only the smallest pores. The procedure described here seems very arbitrary. How do you have any confidence or measure of what soil pores where filled and to what extent?

Response: We appreciate your comment and agree that this detail is more of an assumption.

Planned changes: We will change the verbiage used in abstract to highlight that the "light" water was held under high matric potential by the soil before fully wetted with "heavy" water. We will also add clarification when introducing the experimental design in the methods section 2.1 by explaining that the "light" water applied first to the soil is held under high matric potential being the only water within the soil after removing nearly all other water via oven drying the soil (Adams et al., 2019). We will further explain that we cannot confirm the location of the "light" water within the soil, but that it is likely residing as a thin layer around soil particles and/or within the smallest pores of the perturbed soil due to the adhesion properties of soil particles and the high capillary tension enacted by the smallest pores within the soil. This will work off of the planned

additions to the introduction for the role of soil physics in relation to our study (covered in more detail in response to referee #2 general comment 1).

L118: Are you sure this was the extraction temp? Did you use boiling water? Laramie is pretty high elevation and thus water has a low boiling point.

Response: We appreciate your concerns as the elevation of Laramie would play a factor if we were using boiling water baths for extraction. However, the extraction system at the UW Stable Isotope Facility uses electric heating coils and direct temperature measurements of each extraction vessel using thermistors, thus we are certain of these reported temperatures.

Planned changes: We will add clarification in the methods section about how temperature is controlled and measured using our extraction apparatus.

L122: 95% is still not ALL of the water.

Response: We appreciate your comment and agree that 95% is not all of the water. We were following the guidelines and observations presented in West et al., 2006 where the isotopic signature did not change more than the limitations of instrument accuracy in measuring the stable isotope ratios of water extracted between 95% vs 100%. The majority of our samples had 100% extraction efficiency and the lowest percentage extracted was actually 99%.

Planned changes: We will change this statement to "greater than 99%". In addition, we plan to have additional statement on limitations of CVD and its relation to our study that is covered in more detail in response to referee #2 specific comment L140.

L127: You use the Two Water Worlds terminology here, but you haven't ever really discussed it in any detail in the introduction. I suggest you do so, to help contextualize the rest of the paper.

Response: We appreciate this comment and agree. Please note our response to this concern from referee #2 for planned changes in the introduction, methods, and

discussion (see responses to referee #2's first general comment and specific comment L138).

L136: Three and four hours seems like a long time! On what basis did you choose these times?

Response: We agree these times are long and can be shortened as mentioned in the discussion. These times were used from a preliminary study that determined the time necessary to reach equilibrium for the centrifuge speeds (RPMs) used at which no more water eluted at the given speed. As mentioned in the discussion (line 290-291), these times could be shortened via the support of literature that was found after the preliminary study and lab procedure had taken place.

L138: You never really discuss what is tightly or highly bound, or what the potential mechanisms for this soil water are. There are many aspects to this, from soil pore size, to soil mineralogy, etc. This is a main concept of your paper, but you never give readers any background or basis of understanding how you are using this terminology and "boundness" concepts.

Response: We appreciate this comment and agree that more details should be included on what the tightly bound or matrix water is within the soil.

Planned Changes: With the planned revisions to the introduction for Two Water Worlds and definitions of mobile and matrix water, we will include details on how soil holds onto water at different tensions depending on volumetric wetness and the relation of pore size (likely around line 47 where soil water retention curves are mentioned). Mobile and matrix water will be defined similarly as done by Brantley et al., 2017 where mobile water freely drains or flows under the force of gravity and matrix water is involves hygroscopic and capillary water that does not freely drain or flow under the force of gravity due to the tension enacted by pores and adhesive forces between water molecules and soil particles. We agree "tightly bound fraction of soil water" is not best terminology for this line (line 138) and we will adjust to "a fraction of water held under high tension that

is comprised of capillary and hydroscopic soil water that is rarely directly compared to more mobile waters." Hygroscopic and capillary waters will be defined similar to Brantley et al., 2017 definitions that reference agronomic wilting point as -1.5 MPa: - hydroscopic water is soil water held at high tensions, greater than agronomically defined wilting point, within soil and resides primarily as thin films around soil particles due to strong adhesive forces between water molecules and soil particles - capillary water is soil water held within pore spaces and does not flow freely under the force of gravity and can be held at tensions greater than agronomic wilting point depending on the size of the pore. At high tensions it forms bridges between hygroscopic water films on adjacent soil particles With planned additions covered in response to referee #1 general comment 3, we will explain how these waters are part of a continuum and not discrete fractions of soil water (Sprenger et al., 2018).

L140: Again, are you sure it was ALL of the water left. Or was it 95%? I dont mean to be tedious here, and there are limits to CVD, but that is prwcisely my point. Even CVD at 100 C wont get all the water out that is in interlayers spaces in clays, etc. Are more nuanced discussion is needed (maybe it comes later in the discussion), and at least some acknowledgement of the study's potential limitations is needed. I will look for that as I read. . .

Response: We really appreciate this comment as we think including more details will help tie our work to other recent work on this topic. Planned changes: We will include a statement acknowledging that CVD at 100°C and oven drying soil at 104°C both do not respectively retrieve or remove all of the water from soil as shown by Adams et al., 2019. We will highlight that for the purposes of this study the amount of hydroscopic water that is not retrievable is such a small amount and is likely similar in isotopic composition of the local tap water used for "light" water. Thus, it would have very minimal influence on our results.

L142: I think you should move up the details about the centrifuge and inserts. It hard to envision what you did until you tell us about the inserts.

Response: Thanks for the comment. We agree that some description of the inserts would help at this point.

Planned changes: We will move line 142-149 into the methods section 2.2 after describing the soil and waters used (line 104).

L149: Good that you accounted for evap during the procedure. I assume it was done at room temp, but I could easily see temp being higher inside the centrifuge, especially for 3 to 4 hours. Did you measure this?

Response: We appreciate this comment and see the need for additional details here for readers.

Planned changes: We will include a statement describing that centrifugation was performed with the cooling function on for the Sorvall RC 5B Plus centrifuge and that the internal temperature for the 3-4 hours never exceeded 25°C.

L167: What is atomic fraction? Do you mean isotope ratios (not in delta format)? Or do you mean mixing fraction?

Response: Thank you for this comment as this term was also a concern of referee #1. Please see our detailed response and planned changes to referee #1 general comment 6.

L178: Is this 1% the total mass (water + soil) or just water? If it was total mass, then a decent amount of water lost to evap (and a big shift in isotope ratios) could be contained in the 1% number. I suggest a sensitivity analysis be done to quantify (in isotope terms) what the effects of this much water loss would actually be. It may seem tedious and unnecessary, but with this much handllng of the wet soil, I could easily see evaporation being a bigger factor in isotope results than a casual view would expect.

Response: We appreciate your comment and agree that some clarification is needed. Line 178 refers to 1% mass uncertainty of water, not water + soil. This is why we discounted the impacts of evaporative fractionation on our results.

Planned changes: We will clarify the statement to be 1% mass uncertainty of water.

L217: You state that the BSE waters were not significantly different from the applied waters, but in Figure 3 upper left panel they sure look different to me. It seems that you were not getting back what you put in. This seems problematic.

Response: Thanks for the comment and this is a valid concern. We tried to highlight that this difference is not considered significant, but BSElight water was used for mass balance mixing model due to the shift in values that is in the direction that is typical of fractionation due to evaporation (Allison et al., 1983). This evaporation is likely due to water being applied the low relative humidity of local atmosphere in Laramie, WY since the application by hand of "light" water exposed a lot of surface area of soil to dry atmosphere.

Planned changes: We will add statement in the results that the BSElight isotopic values indicate that likely some evaporative fractionation took place during the application of "light" water recently oven dried soil that had high amount of surface area exposed to dry local atmosphere. Although this changed the isotopic value of water in soil before application of heavy water, the light waters applied and BSElight extracted waters were not significantly different when considered groups (p > 0.05). We will also plan to add ellipses around data points for each group to highlight the multivariate normal distribution of these values at the 95% confidence interval to emphasize that there is overlap between the various waters being investigated.

L218 / Figure 3: Suggest adding A -F labels to the panels in Figure 3. Also, the x-axis labels and ticks seem inadequate.

Response: We appreciate this comment and agree that panel labels and more x-axis labels will help display data.

Planned changes: We will add A-F labels and a description in the figure caption for Figure 3. We will add more ticks on the x-axis labels for d18O for Figure 3.

L222: I suggest keeping the applied water points in all panels in Figure 3 for easier comparison. Perhaps make them dashed outline or ghosted or something to show them but not distract form the time series data.

Response: Thank you for the suggestion as adding the applied waters to all panes would help show how the isotopic signatures of each low-, mid- and high-tension effluents change over time in relation to applied waters.

Planned changes: Points for light and heavy applied waters as well as BSE waters will be added and included in all panels of Figure 3.

L227: So basically, after enough time, all the waters extracted by any means all converged upon the Heavy water signature. And the heavy water signature is the one that you soaked the sample in, but only put a little of the light water in the same samples?

Response: Thanks for bringing up this point. Yes, we inundated the sample in heavy water and applied a relatively small amount of the light water to each sample. Due to the much larger proportion of heavy water applied compared to light water, the data do appear to converge mostly towards the heavy water. So no, they did not converge on heavy water isotope signature in any way inconsistent with mass balance mixing.

Planned changes: We believe by applying the changes mentioned above (in responses and changes to specific comments of referee #2 L217, L218, and L222) that we will help clarify these details. In addition, we will describe in the results that over time the isotopic signatures converged upon the mixture of the two applied waters that has similar isotopic composition of extracted BSElight+heavy and is affected by the large proportion of heavy water applied compared to light water applied, consistent with expectations of mass balance mixing. In the discussion, we will highlight that changing the amount of water applied first and second should be explored in future studies to further understand the degree that antecedent moisture conditions and isotopic ratios created by application of the first water affect the mixing of second applied water and the power of statistical tests.
L231: So, the conclusion is that the samples were all well mixed? OR something else? Because the dont look well mixed to me, especially not until dy 3 or later. Am I missing the point? If so, please explain better.

Response: We appreciate your concerns and hope that by addressing your comment in the previous points (specific comments L217, L218, L222, L227) we can clarify the work. However, we will include some changes to make this section of the results more explicit.

Planned changes: We will add a statement to this part of the results section to explicitly state that these values suggest all water was accounted for in extraction processes and that minimal, if any, fractionation occurred due to evaporation.

L243: How do you evaluate the mixing results if you dont actually known how much of each type of water you put into the soil? Seems to me that with so much more heavy water than light, you are not really evaluating mixing, but more like the time to equilibrium, wherein the heavy water signal just overwhelmed the light because there was so much more of it.

Response: We appreciate your concerns. We agree in the notion that the samples are moving towards equilibrium (as mentioned in line 240 when starting to discuss mixing results), but inherently this would require mixing of two separate water "bodies" until a state of equilibrium is reached. Therefore, both of these terms relate to the process and our discussion should be clear on our use of the terms. Yes, we estimated volume of light water applied using gravimetric water contents, but we did know how much heavy water was applied due to a direct change in mass measurement.

Planned changes: We plan to add to our results that even though all measurements of water additions were not made directly to get a perfect mass balance, the isotope results make sense in that final isotopic value is a weighted mixture of the light and heavy waters. Due to the much larger proportion of heavy water used compared the small amount of light water used, the isotope values are similar to the heavy water, but

are slightly lighter because of the small amount of light water added.

L248: This section reads more like a conclusions paragraph than the start of a discussion. You haven't really supported any of these statements, yet.

Response: Thanks for the comment. We intended here to lay out the structure and direction for the discussion section by highlighting what we feel are the most important take-home points.

L252: What are the proposed mechanisms of mixing? This is hard to determine, because you haven't ever discussed where in soil water is actually held. Is the "mixing" done via diffusion? if so, water self- diffusion in soils is fairly well studied and you could greatly increase the impact of your findings by bringing in some discussion of that work. This seems like an over simplistic analysis of your results, which are a bit fast and loose as it is. No offense intended, just that I am seeking more detail and justification in your measurements and results.

Response: Thanks for the comment. The proposed mechanisms were introduced briefly in the abstract (line 19) and were further discussed later on in the discussion starting line 271. Please see response to referee #2 specific comment L277, which outlines planned changed for more details on self-diffusion.

L266: Are there carbonates in your soil? Easy test with HCl.

Response: Thanks for the suggestion. Indeed, we did test our soil using 1 N HCL and the soil was non-effervescent (i.e. no bubbles formed). In addition, the possible fractionation displayed in d18O data is not explained by isotopic effects due to interaction with carbonates (Meißner et al., 2014) because the shift in isotope values were in the opposite direction than what is predicted due to carbonate interactions.

Planned changes: We will add a statement about our HCl test result in the methods when introducing the soil used in the study, section 2.2.

L270: This is the first time you have acknowledged that your samples have perturbed

soil structure and thus pore sizes. This may be the biggest reason for any isotope effect of any discussed.

Response: We appreciate this comment and agree that the fact that the soil was perturbed needs to be highlighted and acknowledge early in the manuscript. We emphasize however that what we are presenting is a combination approach and we use a fairly artificially disturbed soil for proof of concept. We are unaware of perturbed structure having observed isotope effects on water extracted from soil. We are aware that recent work has highlighted that oven drying soil may change the wettability and surface adhesion of water molecules (Gaj et al., 2019).

Planned changes: We will include a statement in the methods that highlights the fact that soil structure was non-native and perturbed for this study which likely affects the normal distribution of pore sizes. In the planned additional paragraph of the discussion on fractionation effects, we will mention that oven drying the soil may have change the wettability and surface adhesion of water molecules in our study soil (Gaj et al., 2019). We will state that we lacked the capability of identifying these with our lab procedure and that future studies should consider these factors. L277: Finally, the discussion I was yearning for. Can you expand by making some calculations that support these arm waving statements?

Response: Thank you again for your insightful comments. We agree that sharing the equation discussed will help add to the discussion.

Planned changes: We will include the equation for diffusion of solutes in soil that is used for understanding self-diffusion of water within soil as it will be also be useful for helping discuss the conceptual model and details brought up in response to referee #1 general comment 3. We will also demonstrate within the discussion that for a clay soil, the increased porosity affects the tortuosity factor and would therefore affect the rate of diffusion between pores within a clay soil.

L288: Good point on the pore size changing during the spinning.

Response: Thank you!

L290: Yes, shorter spin times!

Response: We agree that this would make the method more realistic for wider applications.

References:

Adams, R.E., Hyodo, A., SantaMaria, T., Wright, C.L., Boutton, T.W., West, J.B., 2019. Bound and mobile soil water isotope ratios are affected by soil texture and mineralogy while extraction method influences their measurement. Hydrol. Process. hyp.13633.

Allison, G.B., Barnes, C.J., Hughes, M.W., 1983. The distribution of deuterium and 18O in dry soils 2. Experimental. J. Hydrol.

Brantley, S.L., Eissenstat, D.M., Marshall, J.A., Evaristo, J., Balogh-Brunstad, Z., Dawson, T.E., McDonnell, J.J., Godsey, S.E., Karwan, D.L., Weathers, K.C., Chadwick, O., Roering, J., Papuga, S.A., 2017. Reviews and syntheses: on the roles trees play in building and plumbing the critical zone. Biogeosciences 14, 5115–5142.

Brooks, J.R., Barnard, H.R., Coulombe, R., McDonnell, J.J., 2010. Ecohydrologic separation of water between trees and streams in a Mediterranean climate. Nat. Geosci. 3, 100–104.

Gaj, M., Lamparter, A., Woche, S.K., Bachmann, J., McDonnell, J.J., Stange, C.F., 2019. The Role of Matric Potential, Solid Interfacial Chemistry, and Wettability on Isotopic Equilibrium Fractionation. Vadose Zo. J. 18.

Meißner, M., Köhler, M., Schwendenmann, L., Hölscher, D., Dyckmans, J., 2014. Soil water uptake by trees using water stable isotopes ($\delta$2H and $\delta$18O)$-$a method test regarding soil moisture, texture and carbonate. Plant Soil 376, 327–335.

Oerter, E.J., Bowen, G., 2017. In situ monitoring of H and O stable isotopes in soil water reveals ecohydrologic dynamics in managed soil systems. Ecohydrology 10,

1–13.

Sprenger, M., Tetzlaff, D., Buttle, J., Laudon, H., Leistert, H., Mitchell, C.P.J., Snelgrove, J., Weiler, M., Soulsby, C., 2018. Measuring and Modeling Stable Isotopes of Mobile and Bulk Soil Water. Vadose Zo. J. 17, 0.

West, A.G., Patrickson, S.J., Ehleringer, J.R., 2006. Water extraction times for plant and soil materials used in stable isotope analysis. Rapid Commun. Mass Spectrom.

---

## Author Response (AR1)

June 4th, 2020

Dear Dr. Josie Geris,

Please find our revised manuscript, point-by-point reply to comments, and marked-up manuscript version for the manuscript *Combination of soil water extraction methods quantifies isotopic mixing of water held at separate tensions in soil*.

We addressed all comments by you and reviewers on the original manuscript by revising and adding clarifications to six main areas/topics within manuscript:

- Time-dependent mixing model
- Soil wetting procedure
- Fractionation effects and offsets in isotope data
- Two Water Worlds hypothesis background and relation to study
- Background on soil physical processes relative to study
- Suggestions for future studies

Alongside addressing each detailed comment, revisions were made throughout the manuscript to maintain consistency and help simplify the main areas/topics listed above. For example, we now define and use more common verbiage for soil water pools (e.g. gravitationally drained water, capillary water, and hygroscopic water) throughout the paper to help relate our approach, methods, and results to other ecohydrological studies. We followed your suggestion to make the conceptual diagram that will help convey these topics and their relation to our analyses and interpretations of results.

During the revision process we found that some of our planned changes were redundant if fully implemented. However, we have highlighted exact changes point-by-point below that address each reviewer comment sufficiently without adding unnecessary redundancies.

Lastly, thank you for the time extension for our minor revisions as the extra time greatly aided in addressing all comments as best as possible.

Sincerely,

William Bowers, Jason Mercer, Mark Pleasants, and David Williams

**Point-by-point reply to reviewer comments in revised manuscript submitted to HESS, *hess-2019-687**

*By William Bowers and co-authors*

This document combines all comments and replies by each reviewer, relative changes in revised manuscript, and marked-up revised manuscript. A detailed reply to each reviewer comment can be found at https://www.hydrol-earth-syst-sci-discuss.net/hess-2019-687/.

For ease of reviewing, we repeat all reviewer comments in **bold font** followed by our changes to the manuscript in blue font noting section in the manuscript where the changes are made. All changes in the marked-up revised manuscript are in underlined dark red font.

Detailed replies:

**Reviewer 1: (1) The manuscript presents the results of an experiment designed to estimate the rate of isotopic mixing in a soil between two waters that differ in their H and O isotope ratios added to soils sequentially following oven-drying. They do show what appears to be a time-dependent process and argue that the time to equilibration is on the order of days (>4 for this soil). I think the manuscript is a contribution to the ongoing and needed effort to better understand the underlying processes that control soil water isotope ratio variation. However, I have what I think are important concerns with the current version.**

**A key, underlying assumption (that the authors acknowledge) is the absence of fractionation effects associated with water addition after oven-drying or with the extraction process. While this may be a valid assumption, there is evidence in their results that it's false, particularly for d18O. The authors assess the potential for enrichment as a function of evaporation by mass balance (comparing mass loss with effluent captures) but this does not account for any fractionation effects associated with clay mineral interactions and is itself subject to errors. It is notable that the quantities used in the mass balance calculations were not the isotope ratios of the added waters, but the value of the isotope ratio of the water extracted by CVD immediately after adding the second water. The authors refer to a "slight" offset, but looking at the data in figure 3, there is apparently as much as a 2‰ difference between the "light" water added and the measured CVD-extracted water. This is not a small difference in my view.**

> Reply: The reviewer referred to fractionation effects present in our study, specifically highlighting that interaction with clay minerals needs to be discussed. Additionally, the reviewer required more justification for using bulk water extracted after "light" water was applied ($BSE_{light}$) for mass balance rather than applied "light" water. We explained the rationale for using $BSE_{light}$ extracted water for mass balance calculations in methods in relation to fractionation offsets reported in previous studies. In addition, the offset is mentioned in results and how light water applied and $BSE_{light}$ extracted water are not statistically different (MANOVA). Lastly, use of $BSE_{light}$ extracted water is covered more in discussion.
> We revised the text with the following statements:
> *L206-213*: The δ values determined for $BSE_{light}$ samples were used in the mass balance model rather than that of the light water added to accommodate for the slight δ offset between these waters. This slight offset may have developed from evaporative fractionation (Allison et al., 1983) that likely occurred when applying the light water to the recently oven-dried soil within the dry local

atmosphere within our lab, or from a small amount of hygroscopic water adsorbed from local atmosphere once soil was removed from the oven (Hillel, 2003). The direction of this slight offset was not consistent with previous observations of isotope effects associated with interactions with clay minerals (Gaj et al., 2017) or carbonates (Meißner et al., 2014).

*L261-265*: The isotope ratio of water recovered using CVD of $BSE_{light}$ samples (bulk sample extraction after light water applied) indicates that potentially the water in the sample at this step was altered slightly by evaporative enrichment of heavy isotopes mixed into the oven dried soil, which had a high amount of surface area exposed to dry local atmosphere. Although this changed the isotopic value of water in soil before application of the heavy water, the light waters applied and $BSE_{light}$ extracted waters were not significantly different ($p > 0.05$).

*L332-336*: We chose to use the isotope value of the bulk water extracted after the light water was applied ($BSE_{light}$) as the end-member in the mass balance model rather than the isotope ratio value of the light water itself. We felt this was justified for the objective of our study, which was to demonstrate the capability of the combined centrifuge-CVD method to evaluate mixing dynamics among different soil water pools.

**Reviewer 1: (2) I am also curious about the method used to add water. The sequence was: oven dry 350g of soil, add 20 ml of "light" water and mix, subsample into centrifuge inserts and immerse in "heavy" water (presumably completely?). These soils were then presumably saturated. Were they allowed to drain at all before centrifugation, etc.? What is the field capacity of this soil and how does it compare to the amount of water added in the first step? I think it would be useful to know if freely-draining water was part of the pool extracted in the first step.**

Reply: More detail has been added to the methods section for the wetting procedure addressing the wetness of samples, drainage during wetting that precludes samples from being completely saturated, and whether low tension extraction contains gravitationally drained water. In addition, we clearly and more broadly applied definitions for common soil water pools in soil near saturation that are incorporated throughout manuscript and highlighted in the updated Fig. 1 and the new conceptual diagram, Fig. 3.
We revised the text with the following statements:
*L130-132*: After the soil cooled from the drying procedure we applied 20ml of the light water with a spray bottle to the 350 g sub-sample and mixed by gloved hands to ensure homogenous application. 18-30g of this slightly wetted soil was gently packed to form soil columns in each of six custom made centrifuge inserts

*L141-143*: the packed inserts were then wetted from the bottom up by immersing in a container with heavy water at a level just below the soil level in each insert. This ensured the soil samples were wetted to near saturation by reducing the chance of air being trapped within the soil matrix.

*L145-147*: Complete saturation was not possible as some water was lost from perforations at the bottom of the inserts when they were removed from the container of heavy water.

[Figure]

*Figure 1*: Soil retention curve for a sandy loam soil using van Genuchten parameters for a general sandy loam (Kosugi et al., 2002). Average volumes (V) from each extraction step of the experiment are illustrated on the right with LT for Low Tension, MT for Mid Tension, and HT for High Tension. Vertical lines are matric potential points of interest: field capacity of -0.033 MPa and agronomic wilting point of -1.5 MPa. The y-axis is effective saturation, a standardized form of volumetric water content. The x-axis has two scales: the top scale is matric potential in MPa and bottom is relative maximum pore size filled at the respective matric potentials (Schjonning, 1992). Samples wetted with both light and heavy waters were near but not at 100% effective saturation.

[Figure]

*Figure 3*: (a) Spatial relationship of the three most commonly discussed water pools that make up the bulk water pool in soil near saturation. Absorbed/hygroscopic water, capillary water and gravity-drained water are depicted in hypothetical cross-section view of two soil particles within the soil matrix. (b) Relative volumes (V) of soil water pools in this study based on Fig. 1 (LT= low tension, MT= mid tension, and HT= high tension) and the relative amount of interactions (size of black arrows) between pools as equilibration time proceeds. (c) Three soil water pools for this study in hypothetical pore space, as diagramed in the first panel, at three equilibration timepoints and various points in the water extraction sequence. Based off of Fig. 1 water extracted at low tension is comprised of gravity-drained water and capillary water, that extracted at mid tension is composed of capillary water, and water extracted at high tension is comprised of capillary water and hygroscopic water. As equilibration time increases, each pool moves closer towards a well-mixed state (i.e. equilibrium).

**Reviewer 1: (3) I think the authors need to more clearly explain their rationale in using the "time-dependent isotope mixing equations." While I see that an exponential fit to the observed data makes sense (at least for d2H) and that there is a tendency for them to converge, I am not sure the logic holds and I think the fitting approach used might obscure the lack of convergence between the CVD data and the centrifuge data (the CVD data plot well below the fitted line in Fig. 4 for d2H on day 7). The idea is that the low –> mid –> CVD represent a gradient from more to less of the recently added "light" water and capture the mixing process as it proceeds. I don't think this approach captures processes that might involve water interacting with clay and I am not convinced that the mixing is "complete" after 4 days based on the results presented in Fig. 4. The authors also acknowledge but do not attempt to explain the very different patterns observed for d18O. I think there's more to these patterns than it taking longer for H218O and H216O isotopologues to mix than those of H. If this were a simple mixing process, shouldn't both H & O behave similarly in terms of trajectory? I think more careful thought needs to go into interpreting these results. I also think the authors should report the clay mineralogy since multiple authors have suggested potential impacts of clay type on extracted water isotope ratios.**

Reply: The reviewer is concerned about the time-dependent mixing model used to evaluate data. We now provide rationale within the methods section for applying the model. The assumption and limitations of the model are now included in the results section and we provide more details about the $\delta^{18}O$ data offset that was highlighted by the reviewer. We expanded our description of the model results in the discussion section relating to the $\delta^{18}O$ values observed in waters extracted at high tension after 7 days of mixing and highlight and explain differences between the time-dependent mixing model and MANOVA.

We revised the text with the following statements:

*L227-228*: We further used a time-dependent isotope mixing equation to approximate the time required for soils to completely mix (i.e. reach equilibrium).

*L299-302*: $\delta^{18}O$ values indicate possible fractionation expressed at day 3 and 7 equilibration timepoints with offsets towards heavier values. Due to these offsets, probability densities were not evaluated with $\delta^{18}O$ data since our time-dependent mixing model did not account for fractionation offsets occurring during equilibration.

*L321-329*: The time-dependent mixing model indicated that complete mixing was achieved at ~4.33 days and this timeframe was consistent with the MANOVA results between the waters recovered at the three tensions. However, at 7 days the waters extracted under high tension were significantly different than those of the BSE$_{light+heavy}$ samples (MANOVA), but were within the 90% credible interval for $\delta^{2}H$ of $\delta_e$ according to the time-dependent mixing model. This highlights a key difference between the statistical methods of comparison: while the MANOVA compares the multivariate normal means across isotopes, our mixing model analysis ignored the $\delta^{18}O$ values due to yet unexplained (see below for further discussion) deviations in the mixing model. Nonetheless, while these methods highlight slight differences in their estimate of when the two added waters were completely mixed across all extracted fractions, they both highlight the long time lags in mixing.

*L337-346*: We observed slightly higher d$^{18}$O values of the extracted water pools at days 3 and 7 than predicted based on simple mixing of the two waters added to the dry soil (Fig. 5). Because we recovered the expected mass of water (>99%) for these samples, we do not feel the observed $^{18}$O enrichment was a result of evaporation. Water interactions with clay minerals (Gaj et al., 2017) and carbonates (Meißner et al., 2014), in contrast, typically result in depletion of $^{18}$O in matrix water. The positive shift in $\delta^{18}O$ of soil water observed in our study however is consistent with

observations reported by Oerter et al. (2014) who found that at low water content d$^{18}$O of matrix water increased in the presence of clays enriched with potassium. We cannot discount the possibility of such ionic interactions in our study. The time course for ionic exchanges with clays that influence the oxygen isotope composition of matrix water might explain why the mixing dynamics observed in our study differed between H and O isotopes. Identifying and analyzing such effects require more thorough analysis.

**Reviewer 1: (4) I was surprised to see the "wilting point" value of -1.5 MPa used. I know the authors are aware that this value is quite high (less negative) compared to values many plants adapted to low-water environments can achieve and experience no damage.**

Reply: We placed into context the term "wilting point" and that this useful reference point is derived for agronomic applications. We now fully explain in the revised methods section our rationale with caveats for use of this reference point in our study.
We revised the text with the following statements:
*L163-168*: We focused on extracting waters near two ecohydrologically relevant pressures for the waters recovered at "low" and "mid" tension: field capacity (i.e., the point at which no more water drains freely under force of gravity) and agronomic wilting point. While field capacity and wilting point varies among different soil types and plants, reference values of 0.033 MPa and 1.5 MPa for field capacity and agronomic wilting point are useful as guidelines for understanding potential boundaries on ecohydrologically separate water pools.

**Reviewer 1: (5) I think the authors should clarify what they mean by "precision" and "accuracy" in the isotope analysis section. Presumably the "accuracy" is some measure of how different the measured/corrected values of an internal reference material were relative to a consensus value, but I think this should be explicitly stated. Similarly, the "precision" is presumably some estimate of variance of the reference material (1 standard deviation of how many replicates?), but again this should be stated.**

Reply: We clarified what we mean by precision and accuracy in the methods section.
We revised the text with the following statements:
*L192-194*: We report the accuracy as the absolute difference between the mean of analyzed lab reference water samples (n=15) and the calibrated value of lab reference water. We report precision as the standard deviation of all lab reference water samples analyzed (n=15).

**Reviewer 1: (6) In line 169 the authors refer to "atomic fraction" when I think they mean "isotope ratio" (e.g., 18O/16O).**

Reply: "Atomic fraction" was changed to "isotope ratio" in the methods section.
We revised the text with the following statements:
*L200-202*:

$$m_{LW} R_{LW} + m_{HW} R_{HW} = m_{LT} R_{LT} + m_{MT} R_{MT} + m_{HT} R_{HT} \tag{2}$$

$m$ is mass of water in kg and $R$ is isotope ratio calculated from either $\delta^2$H or $\delta^{18}$O values for the particular water component.

**Reviewer 2: (1) I found the analysis and discussion of the results to be quite "thin". By that, I mean that there is not an especially in depth or nuanced explanation and discussion of many components**

throughout. **Specific examples follow, but in general, I suggest that the senior authors of the manuscript return to it with a more discriminating eye and identify where it can be "deepened".**

**The authors base the rational for conducting the study on making progress on identifying the potential soil water reservoirs (isotopic or otherwise) that underpin the Ecohydrologic separation, or "Two Water Worlds" (TWW) hypothesis. However, there is only the most minor discussion of this concept in the introduction, and then the authors return to it throughout the results and discussion citing how their findings apply to TWW. This is problematic because the reader doesn't have any firm understating of TWW or how the authors are interpreting TWW (because interps vary). I suggest there be a fuller discussion of TWW and how this study specifically contributes to investigating it in the introduction. Because HESS has an open review process, subsequent reviewers have the advantage of seeing previous reviewer's comments. That is the case here, and while I do not intend to "pile on" the authors, I do support Reviewer #1's comments, especially in regards to the mixing analysis (see my specific comments below).**

Reply: More detailed description of the Two Water Worlds (TWW) hypothesis and the proposed mechanism, ecohydrological separation, to explain the TWW hypothesis have been added to the introduction section. In addition, we have included a more thorough discussion within the introduction section about common soil water pools referred to in ecohydrological separation studies and their relation to the soil water retention curve. The revised manuscript has been adjusted throughout to be consistent with reference to these pools. The relation of soil water pools examined in our study to previous work is also highlighted in a new conceptual diagram, Fig. 3.

We revised the text with the following statements:

*L35-43*: The Two Water Worlds (TWW) hypothesis (McDonnell, 2014) considers that transpiration and runoff to streams derive from separate pools of water that are incompletely mixed in time or across pore regions in the soil. Brooks et al. (2010) presented stable isotope evidence of ecohydrologic separation between plant available water in smaller pore regions and mobile water passing through preferential flow paths when smaller pores were filled, challenging the hypothesis of translatory flow and establishing a mechanism to explain the TWW hypothesis. Yet, most studies examining ecohydrologic separation and the TWW hypothesis fail to differentiate isotopic signatures beyond that of mobile water and bulk soil water. More comprehensive evaluation of soil water isotopes across multiple pore sizes and soil regions is needed to examine recharge processes explaining the TWW hypothesis (Berry et al., 2018; Brantley et al., 2017; Brooks et al., 2010; McDonnell, 2014; Sprenger et al., 2019).

*L47-59*: Characterization of water isotope ratios in soils involves careful consideration of methods used to recover soil water. Depending on the method employed, water is recovered at different energies and the proportion of water extracted is dependent on the volumetric water content of the sample and the soil water retention curve, the relationship between volumetric water content and matric potential (negative equivalent of matric tension) (Sprenger et al., 2015). Terminology for water pools recovered at different applied energies has been debated. For the purposes of relating our study to ecohydrologic separation studies, we define two commonly defined pools, gravity-drained water and matrix water, consistent with recent terminology used by Brantley et al. (2017). Gravity-drained water is the most mobile pool of water within soil that freely drains through large pores under the force of gravity. Whereas matrix water consists of capillary and hygroscopic water that does not drain freely under force of gravity but is held across a broad range of tensions by smaller pores that may or may not be accessible to plants. There is likely a continuum water mobility in soil from the largest pores to the smallest pores with progressively less water mobility as pore size decreases (Sprenger et al., 2018). However, we currently lack methodology to infer the degree of connectivity and dynamics of mixing over time between separate soil water pools extracted at different applied energies.

**Reviewer 2: L44: Need a brief explanation of what in situ equilibration is and some references of papers using either of these methods.**

Reply: We added a reference and more details here.
We revised the text with the following statements:
*L60-62*: Methods to characterize soil water pools in situ include water vapor laser spectroscopy that assumes most mobile soil water is in equilibrium with soil water vapor (Oerter and Bowen, 2017) or field extraction using suction lysimeters (Sprenger et al., 2015).

**Reviewer 2: L100: These waters aren't all that different in isotope compositions. Nota Bene: Kona Deep drinking water is about 0 ‰ in _18O and _2H and is available on Amazon.**

Reply: We included in the methods section our rationale for the selection of the waters used for wetting the soil samples and suggested in the discussion section how future studies should consider selecting waters with greater differences in isotope values.
We revised the text with the following statements:
*L127-129*: We selected these waters because of their contrasting isotopic values representing the natural range expected for cold season (light water) and warm season (heavy water) precipitation in temperate continental interior regions.

*L366-369*: Additional improvements and expanded applications of the combination approach we present should be considered. For example, use of waters with a greater isotopic difference for experimentally wetting dry soil and reversing the order of the addition of the heavy and light waters would better resolve rates of mixing and possible fractionation effects.

**Reviewer 2: L107: The abstract claims that the light water was enough volume to fill only the smallest pores. The procedure described here seems very arbitrary. How do you have any confidence or measure of what soil pores where filled and to what extent?**

Reply: We revised the statement in the abstract to clarify that the first applied water was held at high matric tension. In addition, the relationship between matric tension and pore size is highlighted within revised introduction section (covered above in reply to "Reviewer 2: (1)") and by revised scales on x-axis of Fig. 1.
We revised the text with the following statements:
*L10-13*: We wetted oven-dried, homogenized sandy loam soil first with isotopically "light" water ($\delta^2H$ = -130‰; $\delta^{18}O$ = -17.6‰) to represent antecedent moisture held at high matric tension, and then brought the soil to near saturation with "heavy" water ($\delta^2H$ = -44‰; $\delta^{18}O$ = -7.8‰) representing new input water.

**Reviewer 2: L118: Are you sure this was the extraction temp? Did you use boiling water? Laramie is pretty high elevation and thus water has a low boiling point.**

Reply: We added more clarification in the methods section about the extraction method and equipment for cryogenic vacuum distillation at Stable Isotope Facility of University of Wyoming.
We revised the text with the following statements:
*L152-154*: We performed the CVD procedure at 102°C and <0.1-2.7 Pa vacuum pressure, which were controlled and monitored using heating coils, thermistors, and vacuum gauges. The vacuum pressure used during CVD is not the same as the estimated tension applied using CVD described in section 2.3.

**Reviewer 2: L122: 95% is still not ALL of the water.**

Reply: We adjusted this statement to more accurately reflect the extraction efficiency achieved in our study, rather than simply stating how our efficiencies related to guidelines presented in West et al., 2006. In addition, we highlighted recent work on this topic.
We revised the text with the following statements:
*L154-159*: The final sample masses after extraction were compared to oven-dried masses to determine the recovery of extracted water; every sample processed in our experiment had greater than 99% of water extracted at this step. Recent work has highlighted that CVD near 100ºC or oven drying soil near 105ºC do not extract all of the water from soil (Adams et al., 2019). The amount of water not recovered using CVD in the current study was assumed to be negligible with minimal impact on the isotopic values of extracted water.

**Reviewer 2: L127: You use the Two Water Worlds terminology here, but you haven't ever really discussed it in any detail in the introduction. I suggest you do so, to help contextualize the rest of the paper.**

Reply: We have revised this part of the methods section to be consistent with other changes we have made to the introduction section (covered above in reply to "Reviewer 2: (1)").
We revised the  text with the following statements:
*L166-168*: While field capacity and wilting point varies among different soil types and plants, reference values of 0.033 MPa and 1.5 MPa for field capacity and agronomic wilting point are useful as guidelines for understanding potential boundaries on ecohydrologically separate water pools.

**Reviewer 2: L136: Three and four hours seems like a long time! On what basis did you choose these times?**

Reply: We agree these centrifuge extraction times are long and can be shortened. Our statement addressing this issue (acknowledged in specific comment "Reviewer 2: L290" below) remains in the discussion section for guidance to others who wish to apply the centrifuge method.
The text that is related to this comment:
*L372-374*: Finally, minimizing the time of centrifugation at each step (Fraters et al., 2017) would provide more highly resolved estimates of soil water mixing times and increase sample throughput.

**Reviewer 2: L138: You never really discuss what is tightly or highly bound, or what the potential mechanisms for this soil water are. There are many aspects to this, from soil pore size, to soil mineralogy, etc. This is a main concept of your paper, but you never give readers any background or basis of understanding how you are using this terminology and "boundness" concepts.**

Response: We added more detail and clarification to the introduction section about matric tension and other commonly defined soil water pools (covered above in reply to "Reviewer 2: (1)"). In addition, we modified the methods section with the following text to be consistent with how we define and use these concepts and definitions relating to different pools of soil moisture.
We revised the text with the following statements:
*L178-181*: Afterward, the remining water in in each sample was extracted using CVD and is referenced here as "high tension" extraction; this is a fraction of water held under high tension that is rarely directly compared to more mobile waters within soils that have sufficient volumetric water content to permit sampling with methods like suction lysimeters.

**Reviewer 2: L140: Again, are you sure it was ALL of the water left. Or was it 95%? I dont mean to be tedious here, and there are limits to CVD, but that is prwcisely my point. Even CVD at 100 C wont get all the water out that is in interlayers spaces in clays, etc. Are more nuanced discussion is needed (maybe it comes later in the discussion), and at least some acknowledgement of the study's potential limitations is needed. I will look for that as I read…**

    Reply: We added more clarification and detail to the methods section to address this comment as noted in the reply to "Reviewer 2: L122" above.

**Reviewer 2: L142: I think you should move up the details about the centrifuge and inserts. It hard to envision what you did until you tell us about the inserts.**

    Reply: Details on the centrifuge inserts have been moved up in the methods section as requested by the reviewer.
    We revised the text with the following statements:
    *L130-139*: After the soil cooled from the drying procedure we applied 20ml of the light water with a spray bottle to the 350 g sub-sample and mixed by gloved hands to ensure homogenous application. 18-30g of this slightly wetted soil was gently packed to form soil columns in each of six custom made centrifuge inserts (Fig. 2). The custom steel tube inserts were perforated with small drilled holes at the bottom and fitted with a collar at the top. The collar secured the position of the insert within the sleeve at roughly 19mm above the bottom to establish a reservoir for collecting extracted water through the perforated bottom during extraction by centrifugation (below). We placed four steel screens secured by rubber o-rings at the bottom of each insert to reduce loss of soil yet permit water flow during centrifugation. In addition, we placed a small gravity secured cap on top of each insert to reduce evaporation from soil samples in inserts during equilibration and centrifugation. The caps were loose enough to not generate vacuum within the sample as water was eluted during centrifugation.

**Reviewer 2: L149: Good that you accounted for evap during the procedure. I assume it was done at room temp, but I could easily see temp being higher inside the centrifuge, especially for 3 to 4 hours. Did you measure this?**

    Reply: Temperature during centrifugation was controlled and details have now been added to the methods section.
    We revised the text with the following statements:
    *L162-163*: Centrifugation was performed with the cooling function activated; the internal temperature during centrifugation never exceeded 25ºC.

**Reviewer 2: L167: What is atomic fraction? Do you mean isotope ratios (not in delta format)? Or do you mean mixing fraction?**

    Reply: Changes to manuscript in response to this comment are covered above in reply to "Reviewer 1: (6)".

**Reviewer 2: L178: Is this 1% the total mass (water + soil) or just water? If it was total mass, then a decent amount of water lost to evap (and a big shift in isotope ratios) could be contained in the 1% number. I suggest a sensitivity analysis be done to quantify (in isotope terms) what the effects of this much water loss would actually be. It may seem tedious and unnecessary, but with this much handllng of the wet soil, I could easily see evaporation being a bigger factor in isotope results than a casual view would expect.**

Reply: Clarification was added to the methods section to highlight that 1% mass uncertainty was of water, not water and soil.

We revised the text with the following statements:

*L221-222*: We observed differences of only less than 1% of the mass of the extracted water in all cases, and therefore discounted the impacts of evaporative fractionation on our results and interpretations.

**Reviewer 2: L217: You state that the BSE waters were not significantly different from the applied waters, but in Figure 3 upper left panel they sure look different to me. It seems that you were not getting back what you put in. This seems problematic.**

Reply: We covered the revised changes in response to this comment above in reply to "Reviewer 1: (1)".

**Reviewer 2: L218 / Figure 3: Suggest adding A -F labels to the panels in Figure 3. Also, the x-axis labels and ticks seem inadequate.**

Reply: We agree that adding labels would help. We have updated this figure, which is now Fig. 4 in the revised manuscript.

[Figure]

Figure 4: Isotopic values of water samples in dual-isotope space, $\delta^2 H_{VSMOW}$ (‰) vs. $\delta^{18} O_{VSMOW}$ (‰). (a) Light, Heavy, and BSE(bulk sample extraction) waters with 95% confidence interval ellipses generated by pooled data of Light, Heavy, and BSE waters since the pooled groups were found to be not significantly different with pairwise MANOVA (Table A1 in Appendix) (blue ellipse =

BSE$_{light}$ and Light Water, red ellipse = BSE$_{light+heavy}$ and Heavy Water). (b-f) Waters extracted at low, mid, and high tension for each equilibration timepoint. The 95% confidence interval ellipses from (a) are included in (b-f) for reference.

**Reviewer 2: L222: I suggest keeping the applied water points in all panels in Figure 3 for easier comparison. Perhaps make them dashed outline or ghosted or something to show them but not distract form the time series data.**

Reply: This has been addressed above in reply to "Reviewer 2: L218".

**Reviewer 2: L227: So basically, after enough time, all the waters extracted by any means all converged upon the Heavy water signature. And the heavy water signature is the one that you soaked the sample in, but only put a little of the light water in the same samples?**

Reply: We added clarification in the results section in response to this comment.
We revised the text with the following statements:
*L277-279*: Over time the isotopic ratio values for waters recovered from all three tensions converged upon the expected equilibrium value based on mass balance mixing of the two applied waters, predominantly weighted by the heavy water due to the proportionally much larger amount of heavy water applied.

**Reviewer 2: L231: So, the conclusion is that the samples were all well mixed? OR something else? Because the dont look well mixed to me, especially not until dy 3 or later. Am I missing the point? If so, please explain better.**

Reply: We made some changes to make this section of the results more explicit.
We revised the text with the following statements:
*L286-287*: These values suggest all water applied was accounted for in extraction processes and that minimal, if any, fractionation occurred due to evaporation.

**Reviewer 2: L243: How do you evaluate the mixing results if you dont actually known how much of each type of water you put into the soil? Seems to me that with so much more heavy water than light, you are not really evaluating mixing, but more like the time to equilibrium, wherein the heavy water signal just overwhelmed the light because there was so much more of it.**

Reply: We have clarified within methods section how we performed the calculations that permitted approximation of light water applied and direct measurement of heavy water infused in each sample. In addition, we have incorporated the term equilibrium more throughout the revised manuscript as we agree that two applied waters are mixing overtime to an equilibrium state. This equilibrium state is heavily weighted isotopically by the heavy water since a much larger proportion of heavy water was initially applied to the wetted soil. Please see reply above to "Reviewer 2: L227" for changes to results in relation to this comment.
We revised the text with the following statements:
*L147-148*: Wetted samples were weighed prior to the centrifuge extraction process to determine total wetted weight and amount of heavy water infused in each sample.

*L213-218*:The mass of water remaining in soil samples before high-tension extraction was calculated using gravimetric water contents and the mass of the soil samples after the mid tension centrifuge step. The mass of total water applied to each sample was determined by adding the masses of water remaining in the soil before high tension extraction and water extracted from both

centrifuge steps. The mass of light water applied was determined by subtracting the amount of heavy water infused in the sample (covered in section 2.2) from the mass of total water applied.

**Reviewer 2: L248: This section reads more like a conclusions paragraph than the start of a discussion. You haven't really supported any of these statements, yet.**

Reply: We intended here to lay out the structure and direction for the discussion section by highlighting what we feel are the most important take-home points. No changes were made to the revised manuscript in response to this comment.

**Reviewer 2: L252: What are the proposed mechanisms of mixing? This is hard to determine, because you haven't ever discussed where in soil water is actually held. Is the "mixing" done via diffusion? if so, water self- diffusion in soils is fairly well studied and you could greatly increase the impact of your findings by bringing in some discussion of that work. This seems like an over simplistic analysis of your results, which are a bit fast and loose as it is. No offense intended, just that I am seeking more detail and justification in your measurements and results.**

Reply: The proposed mechanism of mixing, self-diffusion, remains briefly introduced in the abstract with more explicit reference to equilibrium and is now highlighted in more detail in the discussion section. We have added literature references for mechanisms controlling liquid water diffusion rate in soil. We also provide more discussion about how we expect mixing to differ in finer textured and unsaturated soils.
We revised the text with the following statements:
*L16-18*: We assessed differences in the isotopic composition of extracted water over the 7 d equilibration period with a MANOVA and a model quantifying time-dependent isotopic mixing of water towards equilibrium via self-diffusion.

*L347-357*: Since we limited vapor transport and advection in the current study by holding samples in a closed, isothermal vessel near saturation, we assume the isotope mixing among soil pore waters was dominated primarily by self-diffusion of isotopologues by Brownian motion. This mixing towards equilibrium by self-diffusion in hypothetical pore space is shown in Fig. 3. Diffusion rate in soil solution is a function of the diffusion coefficient for the solute of interest, a tortuosity factor, volumetric water content ($\theta$) and the solute effective concentration gradient (Chou et al., 2012). We did not measure these variables in our study; rather we simplified the analysis by lumping these processes into a single empirical parameter ($k$) in our time-dependent mixing model (Eq. (3)). However, we expect soil water content as well as other features that determine tortuosity, like aggregate structure and pore size distribution will have strong influences on the isotopic mixing times of soil water pools. For example, complete mixing in finer textured soils and unsaturated soils will be much longer than those reported here because of these effects, but can be assessed using the general approach we describe.

**Reviewer 2: L266: Are there carbonates in your soil? Easy test with HCl.**

Reply: We did not detect carbonate with 1N HCl and this has been added to the methods section at the point where we introduce the soil used in the experiment.
We revised the text with the following statements:
*L120-121*: We did not detect carbonates in the soil using tests with 1N HCl (Schoeneberger et al., 2012).

**Reviewer 2: L270: This is the first time you have acknowledged that your samples have perturbed soil structure and thus pore sizes. This may be the biggest reason for any isotope effect of any discussed.**

Reply: In response to this comment we have noted in abstract, methods and discussion sections that the structure of the soil used in our study was disturbed and likely lacked complex aggregates. We revised the text with the following statements:

*L18-20*: The simplified and homogenous soil structure and nearly saturated moisture conditions used in our experiment likely facilitated rapid isotope mixing and equilibration among antecedent and new input water.

*L112-115*: We used a sandy loam soil collected from the top 10 cm of the surface from prairie vegetation east of Laramie, WY. Soil was passed through a 2-mm sieve and all coarse litter was removed except for very fine fragments. Our experimental soil therefore was highly homogenized and lacked natural physical structure with complex soil aggregates.

*L318-321*: Complete mixing would likely take longer for undisturbed soil samples with complex aggregate structure compared to our homogenized and disturbed soil samples. The connectivity of water pools within and between soil aggregates and other pore regions for undisturbed soil is likely much lower than in disturbed soils where this complex structure has been reduced.

**Reviewer 2: L277: Finally, the discussion I was yearning for. Can you expand by making some calculations that support these arm waving statements?**

Reply: We added more details to the discussion section about diffusion rate in soils as mentioned above in reply to "Reviewer 2: L252". We found it was more appropriate to reference related work on diffusion rate in soils since we did not take all the necessary measurements to parameterize a more mechanistic model of liquid water diffusion and transport.

**Reviewer 2: L288: Good point on the pore size changing during the spinning.**

Reply: No changes were made in response to this comment.

**Reviewer 2: L290: Yes, shorter spin times!**

Reply: We agree that this would make the method more realistic for wider applications. No changes were made in response to this comment.

[revised manuscript text omitted]